# Using Acceleration Sensors to Diagnose the Operating Condition and to Detect Vibrating Feeder Faults

**DOI:** 10.3390/s25164969

**Published:** 2025-08-11

**Authors:** Leopold Hrabovský, Štěpán Pravda, Robert Brázda, Vojtěch Graf

**Affiliations:** 1Department of Machine and Industrial Design, Faculty of Mechanical Engineering, VSB—Technical University of Ostrava, 17. listopadu 2172/15, Poruba, 708 00 Ostrava, Czech Republic; stepan.pravda@vsb.cz; 2Institute of Transport, Faculty of Mechanical Engineering, VSB—Technical University of Ostrava, 17. listopadu 2172/15, Poruba, 708 00 Ostrava, Czech Republic; robert.brazda@vsb.cz (R.B.); vojtech.graf@vsb.cz (V.G.)

**Keywords:** harmonic oscillation, force sensor, acceleration sensor, electromagnetic exciter, effective vibration velocity

## Abstract

Vibrating feeders are used to empty bulk materials from storage bins, to feed and dispense materials into weighing bins or dispensers, or to feed materials evenly and smoothly into downstream equipment. The harmonic oscillation of the trough can be provided by an electromagnetic oscillator, which consists of an electromagnet consisting of a core and a coil with a given number of coil turns and armature. The aim of this paper has been to verify whether the working condition of the vibrating feeder, i.e., its fault-free operation and the ability to transport the required mass amount of material, can be described on a basis of the measured vibration values using acceleration sensors. This paper describes three experimental methods that allow us with the use of force sensors to measure the adhesion force of the electromagnet and the deformation force of the bent leaf springs, and the use of acceleration sensors to measure the vibration on the trough and on the steel frame of the vibrating feeder. The highest average value of the effective vibration velocity (56.7 mm·s^−1^) in the horizontal plane was measured on a steel frame of a vibrating feeder using FR4 Epoxy leaf springs with a stiffness of 47.8 N·mm^−1^ and a weight of 2.57 kg of conveyed material per trough. The lowest average value of the effective vibration velocity (24.6 mm·s^−1^) has been measured at a weight of 5.099 kg of material conveyed on the trough. We can state that from the analysis of the measured vibration velocities transmitted to the steel frame of the vibrating feeders, it is possible to monitor the partial phases of their operation and diagnose any faults that may occur. It is also possible to determine whether the optimal amount of bulk material is being loaded onto the trough.

## 1. Introduction

Vibrating feeders are short vibrating conveyors with the trough vibration provided by a harmonic vibration source, the so-called vibration exciter [1]. The vibration exciter is usually implemented as a forced drive by a crank mechanism [2,3]; as a drive by a mechanical exciter with unbalances [4,5,6] or as an electromagnetic drive [7,8].

The information, i.e., the electrical signals, detected by the vibration sensors can be used to diagnose the working operation of vibrating feeders in locations that may be in a considerable distance from where the vibrating feeder is installed. Such diagnostics of vibrating feeder parameters can provide controllers in the control centre with the information on whether the required amount of material is on the trough of the vibrating feeder, whether the vibrating feeder is in an optimal operating state or in a fault state.

P. Czubak et al. in their paper [9] analysed the influence of such parameters as the weight of the conveyed material and frequency on the reduction of vibration transmission to the bottom layer. They have come to the conclusion that the choice of optimal parameters, such as accurate calibration of spring stiffness, selection of materials with high absorption capacity and correct sizing of vibration elements, can significantly contribute to reducing vibration transmission to the substrate.

W. Surówka and P. Czubak in their paper [10] described the operational behaviour of vibrating conveyors at resonance and its impact on the efficiency of material conveying.

In vibrating conveyors or feeders, the vibration is considered to be the movement of the trough and other structural parts, individual parts of which oscillate around the equilibrium position. The level of vibration of the trough transmitted to the frame or foundation, to which the frame of the vibrating machines is mechanically attached, is significantly influenced by a number of parameters, the most important of which are the weight of the trough not filled with material, the weight of the conveyed material, the amplitude, and the frequency of the trough’s vibration and the rotor speed of the vibrating motor.

### 1.1. Vibrations—A Diagnostic Indicator of the Operational Status of the Vibratory Feeder

The aim of the measurements carried out in this paper has been to verify the assumption on whether the monitoring of vibrations (using vibrating sensors) transmitted to the supporting frame of a vibrating feeder can provide information about its operating characteristics.

Experimental tests have been carried out using two vibration sensors. One sensor has been installed on the trough of the vibrating feeder, the other on its supporting frame.

The vibrating feeder was equipped with an electromagnetic vibration exciter, the basic part of which consists of an electromagnet, which consists of an armature, a core and a coil with a given number of coil turns. An electromagnet [11] is a coil with a core of magnetically soft steel used to create a temporary magnetic field. The principle consists in transforming energy of the electromagnetic field into the mechanical energy. The magnetic force [12] is generated when an electric current passes through the winding of a coil on a steel core, which attracts a movable part, called the armature. The magnetic flux of the electromagnet and the attractive force of the electromagnet directly depends on the magnitude of the electric current I [A] flowing through the coil, the number of coil turns N_c_ [–] and indirectly on the length δ [m] of the air gap between the core and the armature [13,14,15]. In practice, the attractive force is limited by the total magnetic conductivity of the electromagnet core and the magnetic flux dissipation [16].

### 1.2. Characteristics of the Electromagnet—The Essential Component of the Electromagnetic Exciter

The maximum attractive force F_max_ [N] of the electromagnet [17,18,19] can be determined according to Equation (1), assuming that the permeability of the vacuum μ_0_ [H·m^−1^] (μ_0_ = 1.257 [N·A^−2^]) is known, as well as the magnetic induction in the air gap B_δ_ [T] and the cross-section of the contact area of the electromagnet core S [m^2^].(1)$$F_{max}=\frac{1}{2}\cdot \frac{B_{\delta }^{2}}{\mu_{0}}\cdot S\left[N\right],$$

The cross-section S [m^2^] of the contact surface of the electromagnet core of the electromagnetic exciter of the vibratory feeder described in this paper is specified in Figure 1.

If the distance of the armature δ [m] from the electromagnet core, the number of coil turns N_c_ [−], the current I [A] flowing through the electromagnet coil, the permeability of the vacuum μ_0_ [H·m^−1^] and the contact surface of the electromagnet core S [m^2^] are known, it is possible to express the attractive force F_h_ [N] of the electromagnet by Equation (2).(2)$$F_{h}=\frac{N_{c}^{2}\cdot I^{2}\cdot \mu_{0}\cdot S}{4\cdot \delta^{2}}\left[N\right],$$

The property of a coil is characterized by its inductance L [H]. The inductance is a physical quantity expressing the ability of an electrically conducting body flowing with an electric current to generate a magnetic field in its surroundings. A coil with a larger inductance acts in an AC circuit in such a way that it generates higher resistance to the current. This is due to the fact that the coil induces a voltage directed by its own induction against the voltage of the source. A magnetic field periodically appears and disappears in the coil, so there is no heating of the coil. The action of the coil on the alternating current is characterized by the quantity of inductive reactance (inductance X_L_ [Ω]).

Since the voltage induced in the coil also depends on the rate at which the alternating current changes, it is obvious that the inductance also depends on the frequency of the alternating current. The greater the frequency of the AC current, the greater the inductance of the coil. The coil of inductance L [H] has inductance X_L_ [Ω], in an AC circuit, for which the relation (1) applies.

The trough with a horizontally situated bottom (trough inclination angle β = 0 deg) was mechanically (by means of screw connections) attached to the end parts of four mounted pieces of leaf springs, each L_s_ = 88 mm, obliquely (at an angle α = 30 deg). The solenoid coil of the electromagnetic exciter has been powered from an amplitude/frequency controller (FQ1 DIG Process Controller) [20].

In paper [21] V. Korendiy et al. presented theoretical modelling and experimentally obtained data (defining the influence of design parameters on the operational stability and transport efficiency) for the assessment of dynamic properties of vibrating conveyors.

G. Cieplok in his paper [22] declared by numerical simulations the ability of easier control of undesired resonant vibrations by optimizing the geometry of the vibratory drive parameters.

With AC single-phase electromagnets [23,24] the current is determined by the resistance and self-inductance of the coil, which depends on the position of the armature. If the resistance of the coil is negligible with respect to its reactance, the magnetic flux will be constant and the electromagnet will induce a constant pull at any position of the armature. Resistance, which cannot be neglected, will manifest itself by altering this ideal tensile characteristic. In the initial position, when the air gap is large, the coil impedance is low and the coil draws a high current. The voltage drop induced on the coil resistance will cause the voltage drop across the reactance to generate a significantly weaker magnetic flux [25]. Therefore, the initial thrust will be relatively small. If the air gap decreases, the reactance of the coil increases, the current and voltage drop across the coil resistance decreases, the voltage across the reactance increases, and therefore the thrust increases because the magnetic flux corresponding to the voltage across the reactance gradually increases.

In electromagnetic exciters, the effect of the excitation force is induced by the dynamic force generated during the straight-line reciprocating uniform motion of the metal armature of the electromagnet [26]. The armature of the electromagnet is firmly connected to the trough. The core of the solenoid is connected to the exciter body via pre-tensioned springs.

If the exciter is supplied with AC current at a frequency of 50 Hz, the trough oscillates at a frequency of 100 Hz. If we include a frequency rectifier in the circuit, the oscillation of the exciter is reduced by half and oscillates only at 50 Hz frequency [27].

A simple harmonic motion [28] is the typical motion of a mass on a spring when subjected to a linear elastic reciprocating force given by Hooke’s law. The motion is sinusoidal in time and exhibits a single resonant frequency [29], see Figure 2.(3)$$y\left(t\right)=A\cdot sin\left(\omega \cdot t\right)=A\cdot sin\left(\sqrt{\frac{s_{c}}{m_{2}}}\cdot t\right)\left[m\right],$$

S. Ogonowski and P. Krauze describe in [30] how the dynamic properties of magnetorheological dampers and the motion trajectory of vibration devices can be influenced by the magnetic field control.

The authors M. Pesík and P. Němeček in their paper [31] analysed various types of vibration isolation elements (such as rubber dampers, spring systems and viscous dampers) and evaluate their effectiveness (reduction of vibration transmission to the structural frame and bottom layer) in various operating conditions of vibrating conveyors.

The aim of the paper has been to verify whether the working condition of the vibrating feeder, i.e., its fault-free operation and the ability to transport the required mass amount of material, can be described on a basis of the measured vibration values using acceleration sensors.

The main objective of the realized signal measurements (which define the magnitude of the vibrations in three mutually perpendicular planes) was to determine whether (with varying input values, namely the angle of throw, the angle of inclination of the trough and the use of an electromagnetic vibration motor) it is possible to obtain (from the measured magnitudes of the vibrations acting on the frame of the vibrating conveyor) information about the operating characteristics and the mass of material to be conveyed on the trough with respect to the stiffness of the leaf springs supporting the vibrating masses.

## 2. Materials and Methods

Section 2.1, Section 2.2 and Section 2.3 of this chapter describe the following three laboratory devices that were created to experimentally determine the actual value:-The holding force F_h_ [N] of the electromagnet depending on the distance δ [m] of the armature from the core of the electromagnet;-Stiffness s_cj_ [N·m^−1^] of leaf spring made of material (FR4 Epoxy, steel, plastic PCCF);-Effective trough vibration velocity and effective frame vibration velocity v_*(l,i,j)k,m_ [mm·s^−1^].

These are used in a laboratory vibrating conveyor, the trough oscillation of which is induced by the electromagnetic exciter (see Figure 1).

### 2.1. Laboratory Device for Detecting the Magnitude of the Adhesive Force of an Electromagnet

The structural design of the experimental device, see Figure 3, was designed at the (Department of Machine and Industrial Design, Faculty of Mechanical Engineering, VSB-Technical University of Ostrava in the SolidWorks^®^ software environment Premium 2012 × 64 SP5.0.

The experimental device consists of a frame 4, to the lower part of which the electromagnet 1 is attached by means of screw connections. The armature of the electromagnet 2 is mechanically connected to the force transducer 3 (type MCF30-500 [32]), which is attached to the supporting frame 4. The force attracting the armature of electromagnet 2 (with the parallel lower surface and being 2 mm (or 4 mm) away from the upper surface of the electromagnet) to the electromagnet 1 was detected by force sensor 3, the signal of the measured quantity of the tensile force was displayed in the DEWESoft X2 SP5 software environment [33], which was recorded by the DEWESoft DS-NET measuring apparatus (DEWETech s.r.o., Prague, Czech Republic, Tehovská 1237/25, 100 00 Prague 10) [34], see Figure 4.

The strain gauge cable 3, see Figure 4, terminated with a D-Sub 9-pin plug, has been connected to the DS NET BR4 module [34]. Connectors RJ45 of the network cable are used to connect the module DS GATE [34] to PC (ASUS K72JR-TY131), in which the software DEWESoft X2 SP5 is installed.

### 2.2. Laboratory Device Designed to Determine the Stiffness of a Leaf Spring

The stiffness of the leaf spring s_c_ [N·m^−1^] can be analytically calculated according to relation (4), provided that the modulus of elasticity E_s_ [Pa] of the material, of which the spring is made and the moment of inertia I_x_ [m^4^] are known.(4)$$y_{Mi}=\frac{F_{Mi}\cdot L_{s}^{3}}{3\cdot E_{s}\cdot I_{x}}\to F_{Mi}=y_{Mi}\cdot s_{c}\left[m\right];s_{c}=\frac{3\cdot E_{s}\cdot I_{x}}{L_{s}^{3}}\left[N\cdot m^{-1}\right];I_{x}=\frac{L_{s}\cdot t_{s}^{3}}{12}[m^{4}]$$

The stiffness of the leaf spring (constant of proportionality) s_cj_ [N·mm^−1^] (j = 1 to 10—number of the leaf spring), see Figure 5, used for the vibrating conveyor, is defined as the ratio of the force F_Mijk_ [N] and the vertical displacement y_Mijk_ [mm] of the end portion of the bent spring, which is induced by the force F_Mj_ [N].

The stiffness of 10 pieces of leaf springs s_cj_ [N·mm^−1^], 4 pieces made of FR4 Epoxy (j = 1 to 4); 2 pieces made of steel (j = 5 to 6) and the remaining 4 pieces (j = 7 to 10) have been produced using the FDM 3D printing technology from PCCF filament (Prusament PC Blend Carbon Fiber [35], fill density 15%) printed on a 3D printer (type Pruša MK4S), was experimentally determined from measured values of y_Mk_ [mm] a F_Mijk_ [N] on the laboratory device, see Figure 6. The laboratory device consists of a steel frame 1, to which two positioning tables (linear unit type PT7312-PA) [36] are attached by means of screw connections. A strain gauge load cell 3 (type MCF30-500) is attached to the vertical linear unit 2. The horizontal linear unit 4 with the attached leaf spring 5.

As a result of the rotation of the hand grip of the vertical linear unit 2, the force transducer 3 at a certain moment comes into contact with the end of the leaf spring 5. Further rotation of the hand handle of the vertical linear unit 2 causes the leaf spring 5 to bend, which can be considered as a unilaterally fixed beam in terms of elasticity and strength (see Figure 6c).

A strain gauge force transducer cable (type MCF30-500), terminated with a D-Sub 9-pin plug, has been connected to the DS NET BR4 module [37]. Connectors RJ45 of the network cable are used to connect the module DS GATE [37] to PC (ASUS K72JR-TY131), in which the software DEWESoft X2 SP5 is installed. In the DEWESoft X2 SP5 software environment, the signals of the measured quantity (compressive force F_Mijk_ [N]) were recorded and detected by the DEWESoft DS-NET measuring apparatus [37], see Figure 6d.

### 2.3. Laboratory Model of Vibrating Feeder

Measurement of the effective vibration velocity values v_*(l,i,j)k,m_ [mm·s^−1^] (where *—leaf spring material, l = x, y, z—axes of the coordinate system, i—acceleration sensor number, j—number of measurements, k—amplitude value set on the FQ1 DIGcontroller, m—weight of material on the trough), was carried out on a laboratory model of the vibrating feeder, see Figure 7. The oscillation velocity is used at low and medium frequencies (10 Hz–1000 Hz) to identify faults manifested at these frequencies.

The oscillating motion of trough 1 (its own weight m_z_ = 3.37 kg) is generated by an electromagnetic oscillator, see Figure 7a, which consists of electromagnet 4 (type ME 32/31 K03Z) and the armature 5. The trough 1 is mechanically attached to one end of the leaf springs 3 of rectangular cross-section and to the armature of the electromagnet 5. The second end portion of the leaf springs 3 is attached to the steel frame 2 of the vibrating feeder.

The solenoid coil 4 was supplied with current from the amplitude/frequency controller 6 (FQ1 DIG Process Controller [17]), see Figure 7b. The digital control of the linear feeder using the FQ1 DIG Process Controller 6 (MP Elettronica, Sesto San Giovanni, Italy.) allows the optimization of the vibrating feeder operation by finding its resonant frequency (maximum output), thus eliminating its lengthy and difficult mechanical calibration.

Effective vibration velocity values v_*(l,i,j)k,m_ [mm·s^−1^] of the vibrating trough 1 and steel frame 2 of the vibrating feeder, see Figure 8a, were detected by two acceleration sensors 3 (type PCE KS903.10 [38]).

During the experimental vibration measurements on the vibrating feeder model the signals from the acceleration sensors 3 were recorded by the DEWESsoft SIRIUSi-HS 6xACC, 2xACC+ 5 measuring apparatus [39], see Figure 8b. The time records of the measured values have been transformed by the measuring apparatus into effective values of the broadband velocity. The effective velocity values v_*(l,i,j)k,m_ [mm·s^−1^] of the periodic waveform were displayed on a PC 6 monitor in the DEWESoft X measurement 2025.1 software environment.

The measurement chain, see Figure 8c, presents a sequence of interconnected instruments and devices allowing to detect and process the measured vibration signals of the trough 1 and steel frame 2 of a laboratory model of a vibrating feeder with an electromagnetic vibration exciter.

## 3. Results

### 3.1. Measurement of Electromagnet Holding Force Values

The measured values of the holding force F_h_ [N] of the electromagnet, at an armature distance δ = 2 mm and 4 mm and amplitude A = 10% to 99%, are shown in Table 1.

**Table 1 sensors-25-04969-t001:** Measured size of the holding force F_h_ [N] of the electromagnet, at frequency f = 50 Hz, armature distance δ [m] and amplitude A [m].

δ [mm]	A [m]	F_0_ [N]	F_M_ [N]	F_h_ = F_0_ + |F_M_| [N]	δ [mm]	A [m]	F_0_ [N]	F_M_ [N]	F_h_ = F_0_ + |F_M_| [N]
2	10%	18.2	−44.7	62.9	4	20%	16.6	−17.7	34.3
	20% ^1^	18.3	−44.6	62.9		40% ^3^	16.6	−17.3	33.9
	30%	18.4	−44.8	63.2		60%	16.6	−17.4	34.0
	40%	18.3	−44.5	62.8		80%	16.6	−17.3	33.9
	50% ^2^	18.2	−44.8	63.0		99% ^4^	16.6	−17.2	33.8

^1^ see Figure 9a, ^2^ see Figure 9b, ^3^ see Figure 10a, ^4^ see Figure 10b.

**Figure 9 sensors-25-04969-f009:**
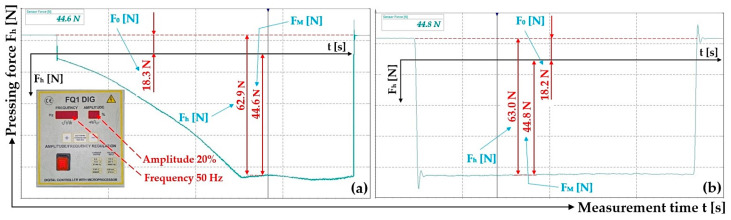
Measured magnitude of the thrust force F_h_ [N] of the electromagnet. Frequency f = 50 Hz, amplitude A [m] (**a**) 20%, (**b**) 50%. Armature clearance δ = 2 mm.

**Figure 10 sensors-25-04969-f010:**
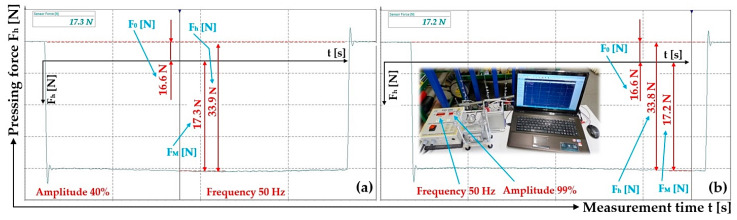
Measured magnitude of the thrust force F_h_ [N] of the electromagnet. Frequency f = 50 Hz, amplitude A [m] (**a**) 40%, (**b**) 99%. Armature clearance δ = 4 mm.

### 3.2. Measurement of the Leaf Spring Deflection

The deflection values y_MK_ [mm] of the end section (determined on laboratory device, see Figure 8) of all 10 pieces of leaf springs 5 are read on the digital display (located at the top) of the vertical linear unit 2. The values of y_MK_ [mm] are recorded in the Table 2 (and the following tables in this Section 2.2).

The deflection y_MK_ [mm] of the end part of the leaf spring 5 is induced by the compressive force (of the vertically moving sliding part of the linear unit 2), the actual magnitude of which F_Mijk_ [N] is measured by the force sensor 3 (type MCF30-500) and recorded in Table 2 (and the following tables in this chapter). From the measured values of the compressive force F_Mijk_ [N] and the deflection y_MK_ [mm], the stiffness s_cijk_ [N·mm^−1^] of a particular (j = 1 to 10) leaf spring is calculated by their ratio.

The deflection values y_Mk_ [mm] of the end section (determined on laboratory device, see Figure 8) of leaf spring No. 1 are shown in Figure 11.

From i = 3 repeated measurements under the same conditions, the values of compressive forces F_Mijk_ [N] and deflection values y_Mk_ [mm] have been measured. In the table of critical values of the Student distribution [40], for the chosen risk α = 5%, the Student’s factor t_α,i_ = 4.3 [40], has been found. According to [40], the standard deviation of the arithmetic mean s_o_ [N·mm^−1^] was calculated for i = 3 repeated measurements. The cardinal error κ_α,i_ [N·mm^−1^], see the penultimate last row of Table 2 and subsequent tables in this chapter, is calculated as the product t_α,i_·s_o_.

The measured values of the pressure forces F_Mi2k_ [N] for the leaf spring made of FR4 Epoxy with a thickness of ts = 2.3 mm are shown in Table 3.

The deflection values y_Mk_ [mm] of the end section of leaf spring No. 2 are shown in Figure 12.

The measured values of the pressure forces F_Mi3k_ [N] for the leaf spring made of FR4 Epoxy with a thickness of t_s_ = 1.9 mm are shown in Table 4.

The measured values of the pressure forces F_Mi4k_ [N] for the leaf spring 4 made of FR4 Epoxy with a thickness of t_s_ = 2.3 mm are shown in Table 5.

The measured values of the pressure forces F_Mi5k_ [N] for the leaf spring made of steel with a thickness of t_s_ = 2.0 mm are shown in Table 6.

The deflection values y_Mk_ [mm] of the end section (determined on laboratory device, see Figure 8) of leaf spring No. 5 and No. 6 are shown in Figure 13.

The measured values of the pressure forces F_M6jk_ [N] for the leaf spring made of steel with a thickness of t_s_ = 2.0 mm are shown in Table 7.

The measured values of the pressure forces F_Mijk_ [N] for the leaf spring No. 7 to 10 made of plastic PCCF with a thickness of t_s_ = 2.0 mm are shown in Table 8.

**Table 8 sensors-25-04969-t008:** Stiffness of leaf spring No. 7 to No. 10 (j = 7 to 10). Thickness t_s_ = 2.0 mm, material—plastic PCCF.

k	j = 7 to 10						
	i	1		2		3	
	y_Mk_ [mm]	F_Mijk_ [N] ^1^	s_cijk_ [N·mm^−1^]	F_Mijk_ [N]	s_cijk_ [N·mm^−1^]	F_Mijk_ [N]	s_cijk_ [N·mm^−1^]
0	0	0	0	0	0	0	0
1	0.5	0.6	1.2	0.6	1.2	0.6	1.2
2	1	1.1	1.1	1.1	1.1	1.1	1.1
3	1.5	1.6	1.1	1.6	1.1	1.6	1.1
4	2	2.1	1.1	2.1	1.1	2.1	1.1
5	2.5	2.5	1.1	2.7	1.1	2.7	1.1
$\bar{s_{cjk}}=\sum_{k=1}^{5}s_{cijk}/k[N\cdot {mm}^{-1}]$			1.1	1.1		1.1	
$\bar{s_{cj}}=\sum_{i=1}^{3}\bar{s_{cijk}}/i[N\cdot {mm}^{-1}]$			1.1				
$\kappa_{5\%,i}[N\cdot {mm}^{-1}]$			0.0				
$s_{cj}=\bar{s_{cj}}+\kappa_{5\%,i}[N\cdot {mm}^{-1}]$			1.1 ± 0.0				

^1^ see Figure 14.

**Figure 14 sensors-25-04969-f014:**
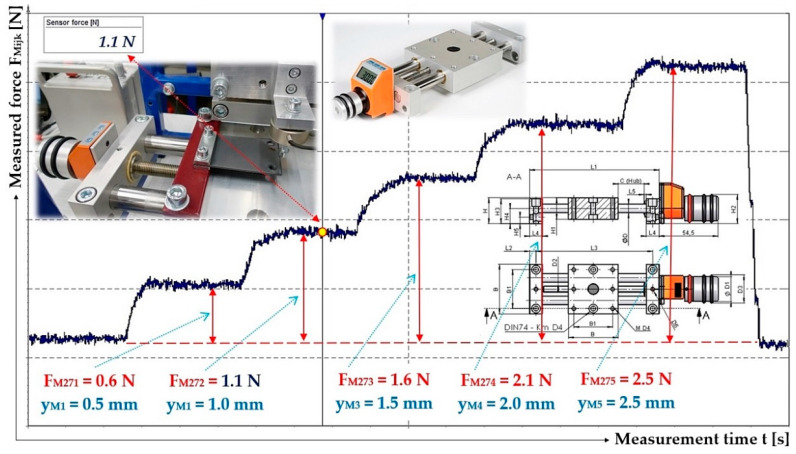
Measured values of the compressive force F_M1jk_ [N], which produces the deflection y_MK_ [mm] of the end part of the leaf spring No. 7 to No. 10, made of plastic PCCF.

### 3.3. Laboratory Device for Determining the Effective Value of the Vibration Velocity

The measurements of the effective vibration velocity values v_c(l,1,j)k,0_ [mm·s^−1^] of the trough (i = 1, see Table 9) and steel frame (i = 2, see Table 9) of the vibrating feeder and the electromagnetic vibration exciter were repeated three times (j = 1 to 3) for each material type (* = c—FR4 Epoxy, s—steel, p—plastic PCCF) from which the leaf springs have been made and set the value k [%] (k = % magnitude of amplitude A [mm]) on the amplitude/frequency controller FQ1 DIG [17]. Effective values of the vibration velocity v_*(l,i,j)k,m_ [mm·s^−1^] of the trough vibration were sensed by two acceleration sensors (PCE KS903.10) and a measuring apparatus (DEWESsoft SIRIUSi-HS 6xACC, 2xACC+ [4,39]) in three mutually perpendicular planes (* = x, y, z—axes of the coordinate system). The measurements were performed in the absence of material on the trough m_q_ = 0 kg, when the electromagnetic exciter only oscillated the trough self-mass m_z_ = 3.37 kg, and at the mass of material on the trough m_q_ = 2.57 kg and 5.099 kg.

Figure 15 indicates the graphical progression (j = 1, see Table 9) of the measured values of the effective vibration velocities v_c(l,i,j)k,m_ [mm·s^−1^] in three mutually perpendicular planes (l = x, y, z) on the trough and on the steel frame of the vibrating feeder, when the trough is supported by four pieces of leaf springs (material FR4 Epoxy, * = c). Amplitude of the trough oscillation A = k·A [mm] (k = 10%), frequency of the trough oscillation f = 50 Hz, mass of the material to be conveyed m = 0 kg.

The 3-times-repeated measurements of the effective values of the vibration velocity v_c(x,i,j)k,2.59_ [mm·s^−1^] of the vibration of the trough (i = 1) and the steel frame (i = 2) of the vibrating feeder with electromagnetic vibration exciter, the trough of which was loaded with the mass of material m = 2.59 kg supported by leaf springs made of material c—FR4 Epoxy, are specified in Table 10.

Figure 16 indicates the graphical progression (j = 1, see Table 10) of the measured values of the effective vibration velocities v_c(l,i,j)k,2.57_ [mm·s^−1^] in the direction of the “x” axis (l = x) on the trough and on the steel frame of the vibrating feeder, when the trough is supported by four pieces of leaf springs (material FR4 Epoxy, * = c). Amplitude of the trough oscillation A = k·A [mm] (k = 20%), frequency of the trough oscillation f = 50 Hz, mass of the material to be conveyed m = 2.57 kg.

The 3-times-repeated measurements of the effective values of the vibration velocity v_c(x,i,j)k,5.099_ [mm·s^−1^] of the vibration of the trough (i = 1) and the steel frame (i = 2) of the vibrating feeder with electromagnetic vibration exciter, the trough of which was loaded with the mass of material m = 5.099 kg supported by leaf springs made of material c—FR4 Epoxy, are specified in Table 11.

Figure 17 indicates the graphical progression (j = 1, see Table 10) of the measured values of the effective vibration velocities v_c(l,i,j)k,5.099_ [mm·s^−1^] in the direction of the “x” axis (l = x) on the trough and on the steel frame of the vibrating feeder, when the trough is supported by four pieces of leaf springs (material FR4 Epoxy, * = c). Amplitude of the trough oscillation A = k·A [mm] (k = 30%), frequency of the trough oscillation f = 50 Hz, mass of the material to be conveyed m = 5.099 kg.

The 3-times-repeated measurements of the effective values of the vibration velocity v_c(x,i,j)k,0_ [mm·s^−1^] of the vibration of the trough (i = 1) and the steel frame (i = 2) of the vibrating feeder with electromagnetic vibration exciter, the trough of which was loaded with the mass of material m = 0 kg supported by leaf springs made of material c—steel, are specified in Table 12.

Figure 18 indicates the graphical progression (j = 2, see Table 12) of the measured values of the effective vibration velocities v_s(l,i,j)k,0_ [mm·s^−1^] in the direction of the “x” axis (l = x) on the trough and on the steel frame of the vibrating feeder, when the trough is supported by four pieces of leaf springs (material—steel, * = s). Amplitude of the trough oscillation A = k·A [mm] (k = 10%), frequency of the trough oscillation f = 50 Hz, mass of the material to be conveyed m = 0 kg.

The 3-times-repeated measurements of the effective values of the vibration velocity v_s(x,i,j)k,2.57_ [mm·s^−1^] of the vibration of the trough (i = 1) and the steel frame (i = 2) of the vibrating feeder with electromagnetic vibration exciter, the trough of which was loaded with the mass of material m = 2.57 kg supported by leaf springs made of material c—steel, are specified in Table 13.

Figure 19 indicates the graphical progression (j = 2, see Table 13) of the measured values of the effective vibration velocities v_s(l,i,j)k,2.57_ [mm·s^−1^] in the direction of the “x” axis (l = x) on the trough and on the steel frame of the vibrating feeder, when the trough is supported by four pieces of leaf springs (material—steel, * = s). Amplitude of the trough oscillation A = k· A [mm] (k = 20%), frequency of the trough oscillation f = 50 Hz, mass of the material to be conveyed m = 2.57 kg.

The 3-times-repeated measurements of the effective values of the vibration velocity v_s(x,i,j)k,5.099_ [mm·s^−1^] of the vibration of the trough (i = 1) and the steel frame (i = 2) of the vibrating feeder with electromagnetic vibration exciter, the trough of which was loaded with the mass of material m = 5.099 kg supported by leaf springs made of material c—steel, are specified in Table 14.

Figure 20 indicates the graphical progression (j = 2, see Table 14) of the measured values of the effective vibration velocities v_s(x,i,j)k,5.099_ [mm·s^−1^] in the direction of the “x” axis (l = x) on the trough and on the steel frame of the vibrating feeder, when the trough is supported by four pieces of leaf springs (material—steel, * = s). Amplitude of the trough oscillation A = k A [mm] (k = 30%), frequency of the trough oscillation f = 50 Hz, mass of the material to be conveyed m = 5.099 kg.

When the vibrating feeder is used with plastic leaf springs, made using the 3D printing method from PCCF filament, the measurements of the effective vibration velocity values v_*(l,i,j)k,m_ [mm·s^−1^] were only carried out for k = 10% to 30% when the conveyed material was not present on the trough surface, and for k = 10% and 20% when the conveyed material was present on the trough (weights of m_q_ = 2.57 kg a 5.099 kg).

The impossibility to perform the measurements at higher k [%] (k = % amplitude magnitude A [mm] set on the amplitude/frequency controller FQ1 DIG [17], see Figure 8b) is due to the low stiffness of the leaf springs s_cj_ [N·mm^−1^] (see Table 8 and Figure 12) printed on a 3D printer from the Prusament PC Blend Carbon Fiber material. Due to the low stiffness of the printed leaf springs (j = 7 to 10, Table 8), the electromagnet armature impacted on the core at k > 30%, which means that the electromagnet adhesion force F_h_ [N] (see Table 1) was multiple times higher than the force F_Mijk_ [N] (see Table 8), which induced a deflection y_Mk_ [mm] of the end parts of the leaf springs. This is higher than the actual distance of the armature δ [mm] from the core of the electromagnet installed as a vibration source in the vibrating feeder with electromagnetic vibration exciter (see Figure 7).

The measured values of effective vibration velocities v_p(x,i,j)k,0_ [mm·s^−1^] of the steel frame (i = 1) and the trough (i = 2), measured along the “x” axis, at a load weight of m = 0 kg and with the leaf spring made of plastic material (* = p), are shown in Table 15.

The 3-times-repeated measurements of the effective values of the vibration velocity v_p(x,i,j)k,2.57_ [mm·s^−1^] of the vibration of the trough (i = 1) and the steel frame (i = 2) of the vibrating feeder with electromagnetic vibration exciter, the trough of which was loaded with the mass of material m = 2.57 kg supported by leaf springs made of material p—plastic, are specified in Table 16.

The 3-times-repeated measurements of the effective values of the vibration velocity v_p(x,i,j)k,5.099_ [mm·s^−1^] of the vibration of the trough (i = 1) and the steel frame (i = 2) of the vibrating feeder with electromagnetic vibration exciter, the trough of which was loaded with the mass of material m = 5.099 kg supported by leaf springs made of material p—plastic, are specified in Table 17.

## 4. Discussion

The vibration of the trough, a vibrating feeder with an electromagnetic vibration exciter, see Figure 7b, supported by leaf springs (made of FR4 Epoxy, steel or plastic PCCF materials) is transmitted through these springs to the steel frame of the vibrating feeder. The magnitude of the vibrations transmitted to the frame of the vibrating feeder varies and depends on the stiffness s_c_ [N·mm^−1^] of the used springs (see Section 2.2 and Section 3.2). If the spring stiffnesses are chosen appropriately, the vibrations transmitted to the vibrating feeder frame are multiple times lower than the vibration of the trough. The vibration of the trough is generated by a source of harmonic vibrations, the so-called vibration exciter. In this paper, an electromagnetic oscillator was used as an exciter for the measuring device (see Section 2.3).

The obtained results confirm the conclusion that it is possible to reduce the transmission of vibrations to the bottom layer in vibrating conveyors using leaf springs, presented in article [41] by J. Michalczyk and P. Czubak.

The measurements of the effective vibration velocities (see Section 3.3) indicate that when using leaf springs made of FR4 Epoxy (optimally chosen stiffness of leaf springs), with a total stiffness of s_cc_ = 47.8 N·mm^−1^, the vibration of the trough is higher in the case of an unloaded trough with conveyed material (the own weight of the trough m_z_ = 3.37 kg), see Table 9 and Figure 21, and lower in the case when there is material on the trough (m_q_ = 2.57 kg or 5.099 kg), see Table 10 and Table 11 and Figure 21.

When using leaf springs made of steel (high stiffness of leaf springs), with a total stiffness of s_cs_ = 107.1 N·mm^−1^, the vibration of the trough is higher when the trough is not loaded with the material being conveyed, see Table 12 and Figure 22, as well as when there is material loaded on the trough (m_q_ = 2.57 kg or 5.099 kg), see Table 13 and Table 14 and Figure 22.

When using plastic PCCF leaf springs (low-stiffness leaf springs), with a total stiffness of s_cp_ = 4.4 N·mm^−1^, the vibration of the trough is also higher when the trough is unloaded with material (trough dead weight m_z_ = 3.37 kg), see Table 15 and Figure 23, and when there is material on the trough (m_q_ = 2.57 kg or 5.099 kg), see Table 16 and Table 17 and Figure 23.

To analyse and justify whether the signals obtained by the acceleration sensors (i.e., the measured effective vibration speeds) can be used to diagnose the working condition of the vibrating feeder, the measured values of the effective spring vibration speeds for k = 40% only, indicated in Table 9, Table 10 and Table 11, will be examined in more detail.

The measured values, when using leaf springs made of FR4 Epoxy and zero filling of the vibrating feeder trough with the conveyed material (m_q_ = 0 kg), indicate that the effective vibration velocity detected on the steel frame of the vibrating feeder reaches the value of 38.5 mm·s^−1^ (see Table 9) and on the trough 65.1 mm·s^−1^. On the steel frame, the effective vibration velocity takes 59.1% of the value of the effective vibration velocity measured on the trough.

With the mass of material to be conveyed on the trough m_q_ = 2.57 kg, both the effective vibration velocities measured on the steel frame and on the trough are the same at 56.8 mm·s^−1^, see Table 10.

With a conveyed material mass on the trough of m_q_ = 5.099 kg, the mean value of the effective vibration velocity measured on the steel frame is 24.6 mm·s^−1^ and on the trough 12.5 mm·s^−1^, see Table 11. On a steel frame, the effective vibration velocity takes 50.8% of the value of the effective vibration velocity measured on the trough.

For leaf springs with optimum stiffness (material FR4 Epoxy—s_cc_ = 47.8 N·mm^−1^), it applies that the mean values of the measured effective vibration velocities take the highest value (65.1 mm·s^−1^ see Table 9) and decrease with an increasing mass of material on the trough (56.8 mm·s^−1^ pro m_q_ = 2.57 kg, see Table 10 and 12.5 mm·s^−1^ for m_q_ = 5.099 kg, see Table 11).

The measured values of the effective vibration velocities, when using leaf springs made of steel, (see Table 12, Table 13 and Table 14) also indicate (according to the mean values of the effective vibration velocities) whether or not the material to be conveyed is on the trough and what mass size it takes on.

On the steel frame, at k = 40% and at zero filling of the trough of the vibrating feeder with the conveyed material (m_q_ = 0 kg), the value of the mean effective vibration velocity was measured to be 10.6 mm·s^−1^ and 18.8 mm·s^−1^ on the trough, respectively, using steel springs. Thus, on the steel frame, the effective vibration velocity is 56.4% of the effective vibration velocity measured on the trough.

With the weight of the material conveyed on the trough m_q_ = 2.57 kg, the mean value of the effective vibration velocity measured on the steel frame is 93.4% of the value of the effective vibration velocity measured on the trough.

With the weight of the material conveyed on the trough m_q_ = 5.099 kg, the mean value of the effective vibration velocity measured on the steel frame is 37.4% of the value of the effective vibration velocity measured on the trough.

For leaf springs that have high stiffness (material—steel—s_cs_ = 107.1 N·mm^−1^), it applies that the mean values of the measured effective vibration velocities are the lowest (18.8 mm·s^−1^ see Table 12) in the absence of the conveyed material on the trough and increase with increasing weight of the material on the trough (19.6 mm·s^−1^ pro m_q_ = 2.57 kg, see Table 13 and 19.0 mm·s^−1^ for m_q_ = 5.099 kg, see Table 14).

Measured values of the effective vibration velocities when using leaf springs made of plastic PCCF (see Table 15, Table 16 and Table 17) also indicate that it can be predicted from the measured values of the effective vibration velocities whether or not there is material (and if so and of what mass size) to be conveyed on the trough of the vibrating feeder.

During the measurements carried out on the laboratory device, i.e., a vibrating feeder with an electromagnetic vibration exciter, using low-stiffness leaf springs (s_cp_ = 4.4 N·mm^−1^), it was not possible to set the amplitude/frequency controller—FQ1 DIG Process Controller [17] to higher values of k [%] than 20%. This was due to the low stiffness of the leaf springs (made of plastic PCCF), which caused the armature to come into contact with the core of the electromagnet—a situation that is unacceptable in electromagnetic exciters used in practice.

On the steel frame, at k = 20% and with zero filling of the vibrating feeder trough with the conveyed material (m_q_ = 0 kg), the value of the mean effective vibration velocity has been measured to be 31.2 mm·s^−1^ and 19.0 mm·s^−1^ at the trough. Thus, on a steel frame, the effective vibration velocity reaches 60.9% of the effective vibration velocity measured on the trough.

With the weight of the material conveyed on the trough m_q_ = 2.57 kg, the mean value of the effective vibration velocity measured on the steel frame is 97.2% of the value of the effective vibration velocity measured on the trough.

With the weight of the material conveyed on the trough m_q_ = 5.099 kg, the mean value of the effective vibration velocity measured on the steel frame is 62.2% of the value of the effective vibration velocity measured on the trough.

Based on the above conclusions of the measured values of the effective vibration velocities (see Table 9, Table 10, Table 11, Table 12, Table 13, Table 14, Table 15, Table 16 and Table 17 and Figure 21, Figure 22 and Figure 23), it can be predicted with a high degree of probability that it is possible to estimate whether or not there is material being conveyed on the trough of the vibrating feeder and also that there is a larger or smaller mass quantity of material on the trough.

Conclusion: From the measurements carried out in the article [4] by L. Hrabovský et al., it can be traced that with inappropriately selected stiffnesses of springs supporting the trough of the vibrating conveyor, it can be traced from the analysis of vibration signals (transmitted to the machine frame) generated by sensors that the vibration of a loaded trough (m_m_ > 0 kg) is higher than the vibration of an unloaded trough (m_m_ = 0 kg) of the vibrating conveyor.

From the analysis of the measured values presented in Section 3.3 it can be concluded that by monitoring the vibrations (using vibration sensors) transmitted to the supporting frame of the vibrating feeder, information about its working properties can be obtained, thus confirming the intended objective of the conducted experiments.

The information, i.e., the electrical signals detected by the vibration sensors, can be used to diagnose the working operation of vibrating feeders in locations that may be in a considerable distance from where the vibrating feeder is installed. Such diagnostics of vibrating feeder parameters can provide controllers in the control centre with the information on whether the required amount of material is on the trough of the vibrating feeder, whether the vibrating feeder is in an optimal operating state or in a fault state.

## 5. Conclusions

In the presented paper, the tables indicate the effective vibration velocity values obtained by measurements, detected by acceleration sensors, on the trough surface and on the frame of the vibrating feeder model with electromagnetic vibration exciter. The trough of the vibrating conveyor is supported by three types of leaf springs, which differ in terms of their stiffness (spring characteristics). The harmonic oscillation of the trough is induced by an electromagnetic oscillator, the frequency and amplitude of oscillation of which is controlled by the controller (amplitude/frequency regulator—FQ1 DIG Process Controller).

The main objective of the realized signal measurements (which define the magnitude of the vibrations in three mutually perpendicular planes) was to determine whether (with varying input values, namely the amplitude of vibrations, the mass of the conveyed material) it is possible to obtain (from the measured magnitudes of the vibrations acting on the frame of the vibrating conveyor) information about the operating characteristics and the mass of material to be conveyed on the trough with respect to the stiffness of the rubber springs supporting the vibrating masses.

Acceleration magnitudes of the effective vibration velocity values measured by sensors have demonstrated and confirmed that if springs of a particular stiffness supporting the vibrating trough are selected appropriately, it is possible to monitor the correct operational operation of the vibratory conveyor and to have information that the required mass quantity of conveyed/sorted material is on the trough of the vibratory machine.

Knowing the magnitude of the vibrations acting on the frame of a particular vibrating feeder (obtained by sensor measurements), with known values of the stiffness of the springs supporting the trough, it is also possible to trace the failure state of their working activities, or to obtain information about the failure or damage of the leaf springs (used on the vibrating machine).

Signals indicating the magnitude of the vibration values acting on the frame of vibrating conveyors/sorting machines, transmitted to the control station, allow monitoring of the operation of vibrating machines at any time without the need for physical inspection of these devices by authorized persons at a place of their installation.

The obtained data on the magnitudes of the measured signals detected by the vibration sensors allowed us to confirm the correctness of the initial idea that (with appropriately designed machine parts) it is possible to monitor the proper working operation and the failure state of vibrating conveyors under operating conditions.

The current trend towards digitalization and computer-controlled or monitored optimum operation of conveyor handling equipment (including vibrating feeders) heavily relies on sensors, measuring equipment and digital signal transmission over any distance.

## Figures and Tables

**Figure 1 sensors-25-04969-f001:**
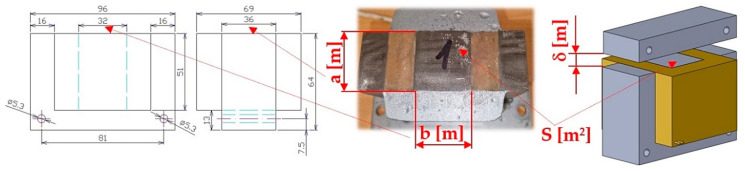
Dimensional sketch of the electromagnet, a/b [m]—length/width of the electromagnet core, δ [m]—armature clearance (air gap).

**Figure 2 sensors-25-04969-f002:**
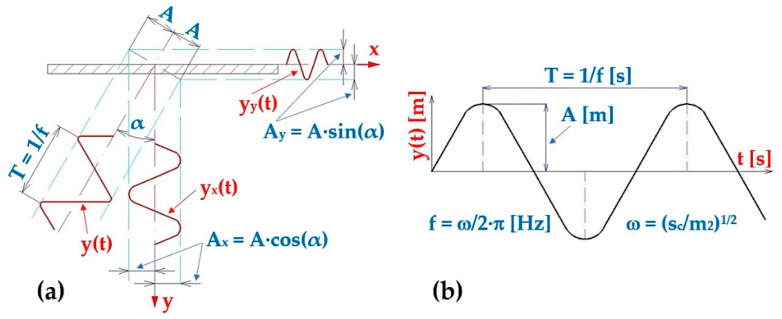
Harmonic oscillation of the vibrating conveyor trough, (**a**) components y_x_(t) [m] a y_y_(t) [m] of the trough deflection, (**b**) harmonic oscillation parameters.

**Figure 3 sensors-25-04969-f003:**
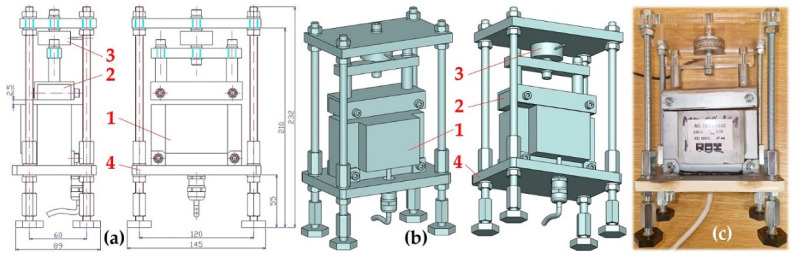
Laboratory device for detecting the magnitude of the electromagnet’s holding force: (**a**) 2D dimensional sketch, (**b**) 3D model, (**c**) implementation. 1—electromagnet, 2—electromagnet armature, 3—force sensor, 4—support frame.

**Figure 4 sensors-25-04969-f004:**
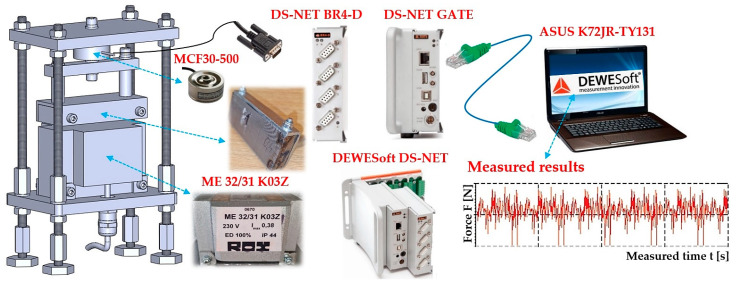
Measuring chain—a sequence of interconnected instruments and devices that enable the detection and processing of measured signals.

**Figure 5 sensors-25-04969-f005:**
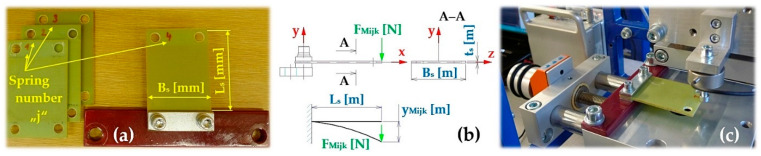
(**a**) leaf springs made of FR4 Epoxy, (**b**) deflection y_Mijk_ [m] of a leaf spring of length L_s_ [m] loaded with force F_Mijk_ [N], (**c**) measuring the deflection of the leaf spring on laboratory device.

**Figure 6 sensors-25-04969-f006:**
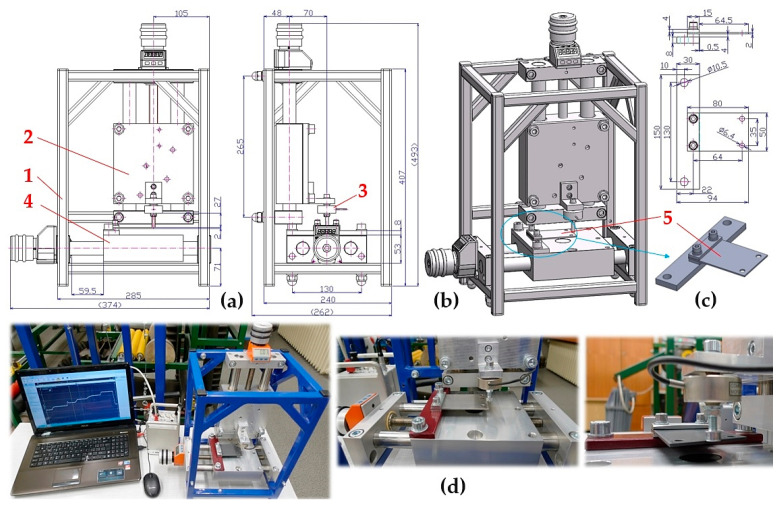
Laboratory device for measuring the stiffness of leaf springs, (**a**) dimensional sketch, (**b**) 3D model, (**c**) attachment of the leaf spring to the flat bar by a bolted connection, (**d**) implemented device. 1—steel frame, 2, 4—positioning table PT7312-PA, 3—MCF30-500 force transducer, 5—leaf spring.

**Figure 7 sensors-25-04969-f007:**
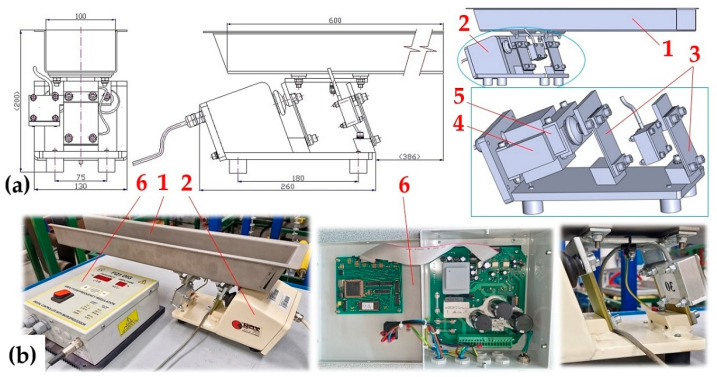
The vibrating feeder with electromagnetic vibration exciter (**a**) dimensional sketch and 3D model, (**b**) implemented measuring device. 1—steel trough, 2—steel frame, 3—leaf spring, 4—electromagnet, 5—armature of the electromagnet, 6—amplitude/frequency regulator—FQ1 DIG.

**Figure 8 sensors-25-04969-f008:**
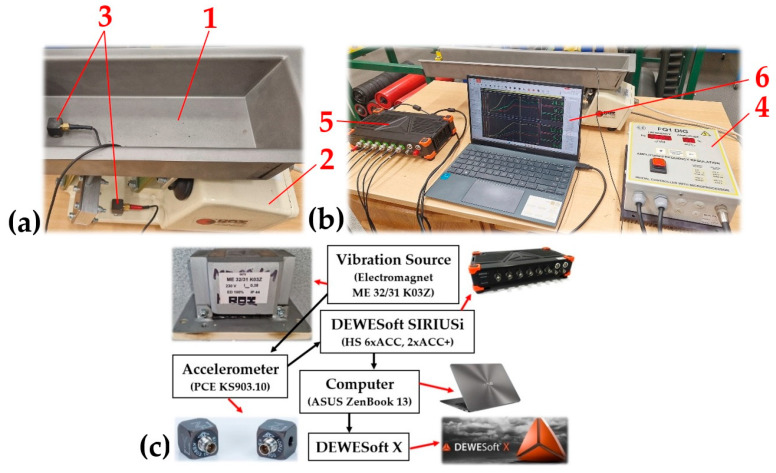
The vibrating feeder with electromagnetic vibration exciter. (**a**) placement of sensors on the trough and the feeder frame, (**b**) control unit and display computer, (**c**) measuring chain. 1—steel trough, 2—steel frame, 3—acceleration sensors, 4—amplitude/frequency controller, 5—measuring apparatus, 6—PC with DEWESoft X software.

**Figure 11 sensors-25-04969-f011:**
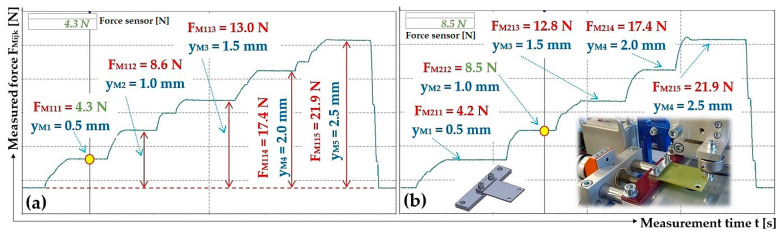
Measured values of the compressive force (**a**) F_M11k_ [N], (**b**) F_M21k_ [N], which produces the deflection y_MK_ [mm] of the end part of the leaf spring No. 1, made of FR4 Epoxy.

**Figure 12 sensors-25-04969-f012:**
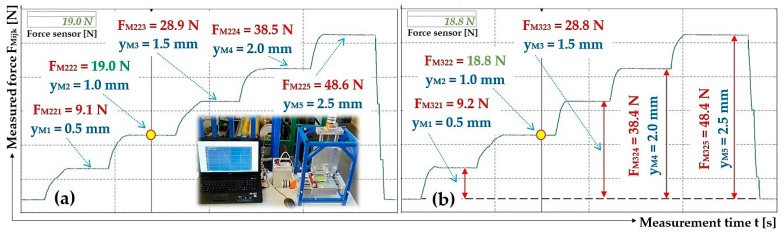
Measured values of the compressive force (**a**) F_M22k_ [N] and (**b**) F_M32k_ [N], which produces the deflection y_MK_ [mm] of the end part of the leaf spring No. 2, made of FR4 Epoxy.

**Figure 13 sensors-25-04969-f013:**
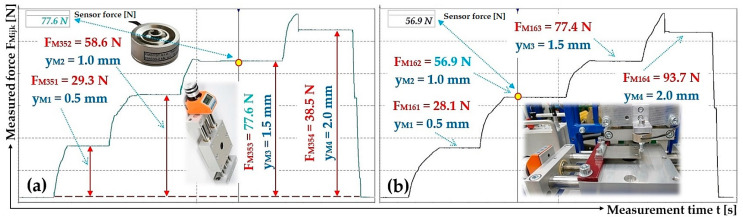
Measured values of the compressive force (**a**) F_M35k_ [N] and (**b**) F_M16k_ [N], which produces the deflection y_MK_ [mm] of the end part of the leaf spring No. 5 and No. 6, made of Steel.

**Figure 15 sensors-25-04969-f015:**
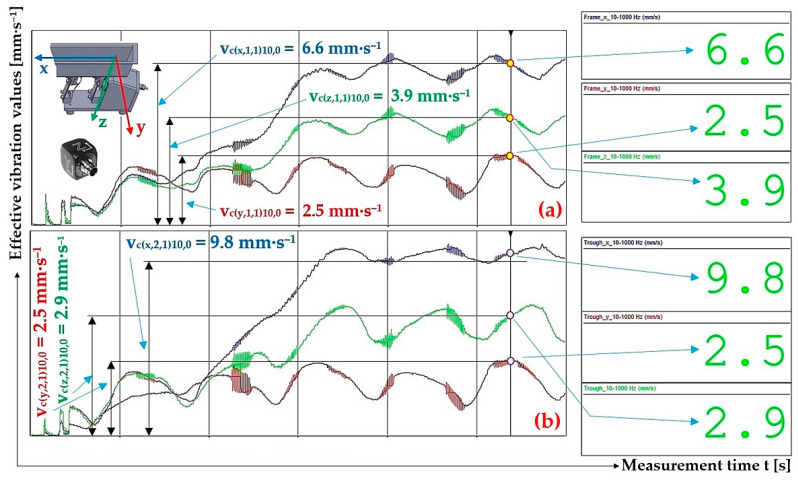
Measured values of effective vibration velocities v_c(l,i,1)10,0_ [mm·s^−1^] in three mutually perpendicular planes of (**a**) trough supported by 4 FR4 Epoxy leaf springs, (**b**) steel frame of vibrating feeder.

**Figure 16 sensors-25-04969-f016:**
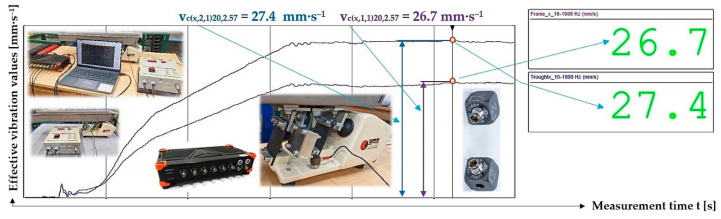
Measured values of the effective vibration velocity v_c(x,i,1)20,2.57_ [mm·s^−1^] in the direction of the x-axis of the trough (supported by 4 FR4 Epoxy leaf springs) and the steel frame of the vibrating feeder.

**Figure 17 sensors-25-04969-f017:**
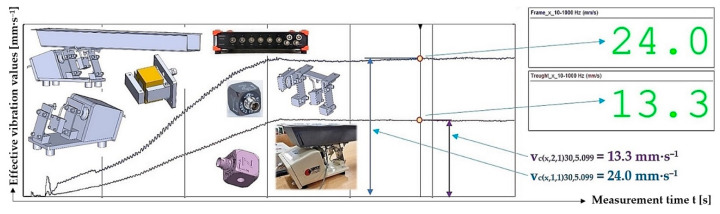
Measured values of the effective vibration velocity v_c(x,i,1)30,5.099_ [mm·s^−1^] in the direction of the x-axis of the trough (supported by 4 FR4 Epoxy leaf springs) and the steel frame of the vibrating feeder.

**Figure 18 sensors-25-04969-f018:**
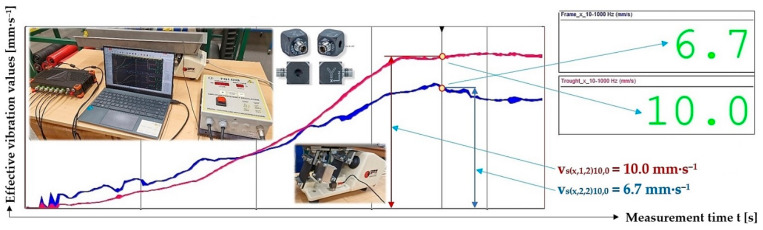
Measured values of the effective vibration velocity v_s(x,i,2)10,0_ [mm·s^−1^] in the direction of the x-axis of the trough (supported by 2 steel leaf springs) and the steel frame of the vibrating feeder.

**Figure 19 sensors-25-04969-f019:**
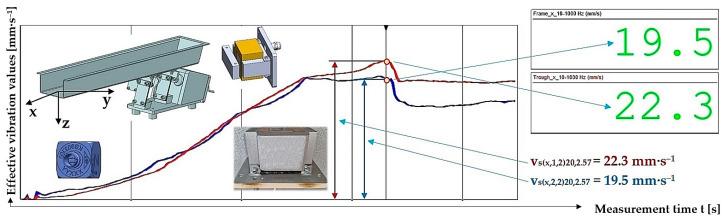
Measured values of the effective vibration velocity v_s(x,i,2)20,2.57_ [mm·s^−1^] in the direction of the x-axis of the trough (supported by 2 steel leaf springs) and the steel frame of the vibrating feeder.

**Figure 20 sensors-25-04969-f020:**
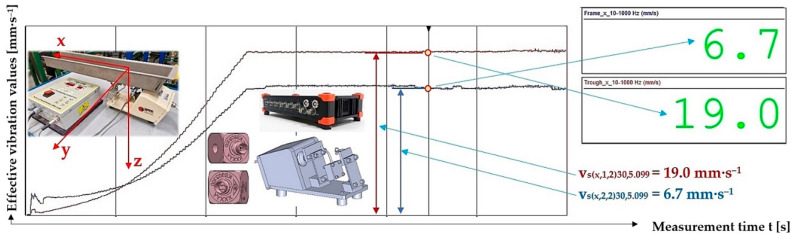
Measured values of the effective vibration velocity v_s(x,i,2)30,5.099_ [mm·s^−1^] in the direction of the x-axis of the trough (supported by 2 steel leaf springs) and the steel frame of the vibrating feeder.

**Figure 21 sensors-25-04969-f021:**
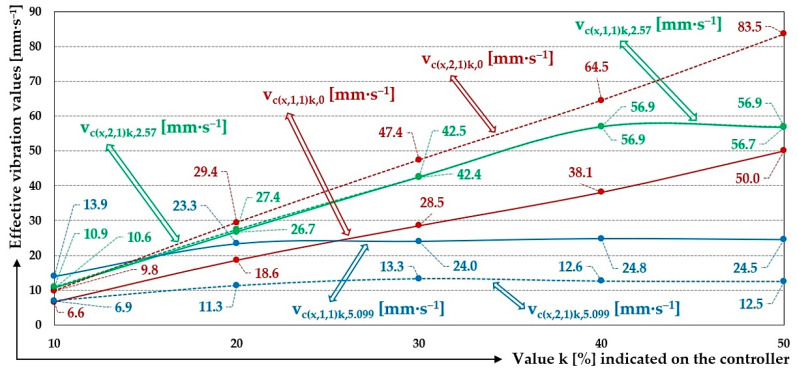
Measured values of the effective vibration velocities v_c(x,i,j)k,m_ [mm·s^−1^] of the oscillation of the steel frame (continuous curve) and trough (dashed curve) of the vibrating feeder with various values of k [%] and the selected material of the leaf springs FR4 Epoxy.

**Figure 22 sensors-25-04969-f022:**
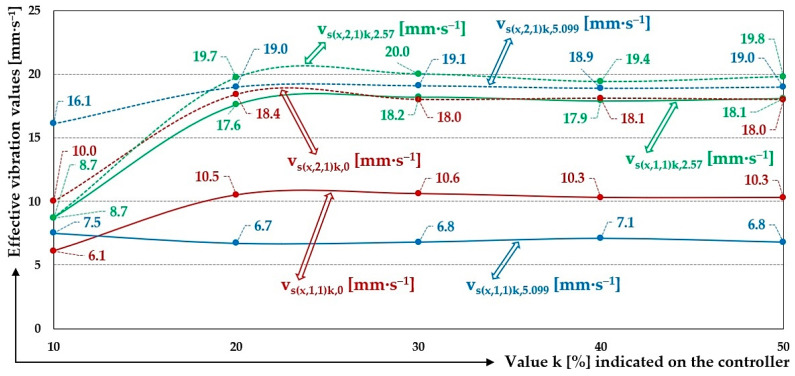
Measured values of the effective vibration velocities v_s(x,i,j)k,m_ [mm·s^−1^] of the oscillation of the steel frame (continuous curve) and trough (dashed curve) of the vibrating feeder with various values of k [%] and the selected material of the leaf springs steel.

**Figure 23 sensors-25-04969-f023:**
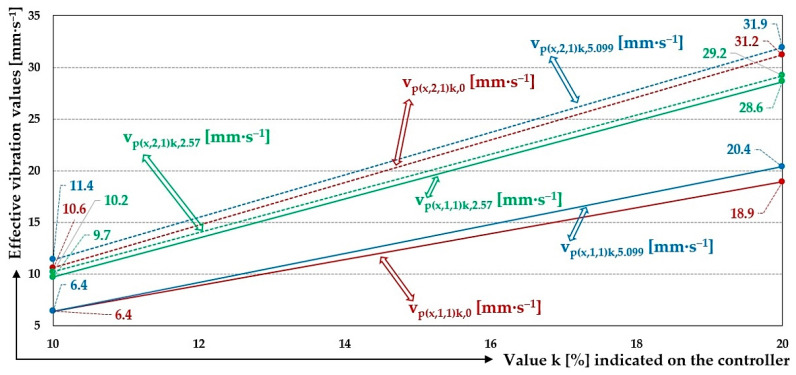
Measured values of the effective vibration velocities v_p(x,i,j)k,m_ [mm·s^−1^] of the oscillation of the steel frame (continuous curve) and trough (dashed curve) of the vibrating feeder with various values of k [%] and the selected material of the leaf springs Plastic PCCF.

**Table 2 sensors-25-04969-t002:** Stiffness of leaf spring No. 1 (j = 1). Thickness t_s_ = 1.9 mm, material—FR4 Epoxy.

k	j = 1						
	i	1		2		3	
	y_Mk_ [mm]	F_Mijk_ [N] ^1^	s_cijk_ [N·mm^−1^]	F_Mijk_ [N] ^2^	s_cijk_ [N·mm^−1^]	F_Mijk_ [N]	s_cijk_ [N·mm^−1^]
0	0	0	0	0	0	0	0
1	0.5	4.3	8.6	4.2	8.4	4.2	8.4
2	1	8.6	8.6	8.5	8.5	8.4	8.4
3	1.5	13.0	8.7	12.8	8.5	12.8	8.5
4	2	17.4	8.7	17.4	8.7	17.3	8.7
5	2.5	21.9	8.8	21.9	8.8	21.9	8.8
$\bar{s_{cjk}}=\sum_{k=1}^{5}s_{cijk}/k[N\cdot {mm}^{-1}]$			8.7	8.6		8.5	
$\bar{s_{cj}}=\sum_{i=1}^{3}\bar{s_{cijk}}/i[N\cdot {mm}^{-1}]$			8.6				
$\kappa_{5\%,i}[N\cdot {mm}^{-1}]$			0.2				
$s_{cj}=\bar{s_{cj}}+\kappa_{5\%,i}[N\cdot {mm}^{-1}]$			8.6 ± 0.2				

^1^ see Figure 11a, ^2^ see Figure 11b.

**Table 3 sensors-25-04969-t003:** Stiffness of leaf spring No. 2 (j = 2). Thickness t_s_ = 2.3 mm, material—FR4 Epoxy.

k	j = 2						
	i	1		2		3	
	y_Mk_ [mm]	F_Mijk_ [N]	s_cijk_ [N·mm^−1^]	F_Mijk_ [N] ^1^	s_cijk_ [N·mm^−1^]	F_Mijk_ [N] ^2^	s_cijk_ [N·mm^−1^]
0	0	0	0	0	0	0	0
1	0.5	9.5	19.0	9.1	18.2	9.2	18.4
2	1	19.5	19.5	19.0	19.0	18.8	18.8
3	1.5	29.1	19.4	28.9	19.3	28.8	19.2
4	2	38.7	19.4	38.5	19.3	38.4	19.2
5	2.5	48.5	19.4	48.6	19.4	48.4	19.4
$\bar{s_{cjk}}=\sum_{k=1}^{5}s_{cijk}/k[N\cdot {mm}^{-1}]$			19.3	19.0		19.0	
$\bar{s_{cj}}=\sum_{i=1}^{3}\bar{s_{cijk}}/i[N\cdot {mm}^{-1}]$			19.1				
$\kappa_{5\%,i}[N\cdot {mm}^{-1}]$			0.3				
$s_{cj}=\bar{s_{cj}}+\kappa_{5\%,i}[N\cdot {mm}^{-1}]$			19.1 ± 0.3				

^1^ see Figure 12a, ^2^ see Figure 12b.

**Table 4 sensors-25-04969-t004:** Stiffness of leaf spring No. 3 (j = 3). Thickness t_s_ = 1.9 mm, material—FR4 Epoxy.

k	j = 3						
	i	1		2		3	
	y_Mk_ [mm]	F_Mijk_ [N]	s_cijk_ [N·mm^−1^]	F_Mijk_ [N]	s_cijk_ [N·mm^−1^]	F_Mijk_ [N]	s_cijk_ [N·mm^−1^]
0	0	0	0	0	0	0	0
1	0.5	3.2	6.4	3.1	6.2	3.1	6.2
2	1	6.5	6.5	6.4	6.4	6.4	6.4
3	1.5	9.7	6.5	9.7	6.5	9.7	6.5
4	2	13.0	6.5	13.0	6.5	13.1	6.6
5	2.5	16.4	6.6	16.5	6.6	16.5	6.6
$\bar{s_{cjk}}=\sum_{k=1}^{5}s_{cijk}/k[N\cdot {mm}^{-1}]$			6.5	6.4		6.4	
$\bar{s_{cj}}=\sum_{i=1}^{3}\bar{s_{cijk}}/i[N\cdot {mm}^{-1}]$			6.4				
$\kappa_{5\%,i}[N\cdot {mm}^{-1}]$			0.1.3				
$s_{cj}=\bar{s_{cj}}+\kappa_{5\%,i}[N\cdot {mm}^{-1}]$			6.4 ± 0.1				

**Table 5 sensors-25-04969-t005:** Stiffness of leaf spring No. 4 (j = 4). Thickness t_s_ = 2.3 mm, material—FR4 Epoxy.

k	j = 4						
	i	1		2		3	
	y_Mk_ [mm]	F_Mijk_ [N]	s_cijk_ [N·mm^−1^]	F_Mijk_ [N]	s_cijk_ [N·mm^−1^]	F_Mijk_ [N]	s_cijk_ [N·mm^−1^]
0	0	0	0	0	0	0	0
1	0.5	6.8	13.6	6.5	13.0	6.5	13.0
2	1	13.9	13.9	13.6	13.6	13.6	13.6
3	1.5	20.8	13.9	20.7	13.8	20.7	13.8
4	2	27.9	14.0	27.7	13.9	27.7	13.9
5	2.5	34.9	14.0	34.8	13.9	35.0	14.0
$\bar{s_{cjk}}=\sum_{k=1}^{5}s_{cijk}/k[N\cdot {mm}^{-1}]$			13.9	13.6		13.7	
$\bar{s_{cj}}=\sum_{i=1}^{3}\bar{s_{cijk}}/i[N\cdot {mm}^{-1}]$			13.7				
$\kappa_{5\%,i}[N\cdot {mm}^{1}]$			0.3				
$s_{cj}=\bar{s_{cj}}+\kappa_{5\%,i}[N\cdot {mm}^{-1}]$			13.7 ± 0.3				

**Table 6 sensors-25-04969-t006:** Stiffness of leaf spring No. 5 (j = 5). Thickness t_s_ = 2.0 mm, material—Steel.

k	j = 5						
	i	1		2		3	
	y_Mk_ [mm]	F_Mijk_ [N]	s_cijk_ [N·mm^−1^]	F_Mijk_ [N]	s_cijk_ [N·mm^−1^]	F_Mijk_ [N] ^1^	s_cijk_ [N·mm^−1^]
0	0	0	0	0	0	0	0
1	0.5	29.9	59.8	29	58.0	29.3	58.6
2	1	57.4	57.4	57.6	57.6	58.6	58.6
3	1.5	76.7	51.1	77.5	51.7	77.6	51.7
4	2	98.1	49.1	97.4	48.7	95.3	47.7
$\bar{s_{cjk}}=\sum_{k=1}^{5}s_{cijk}/k[N\cdot {mm}^{-1}]$			54.3	54.0		54.1	
$\bar{s_{cj}}=\sum_{i=1}^{3}\bar{s_{cijk}}/i[N\cdot {mm}^{-1}]$			54.1				
$\kappa_{5\%,i}[N\cdot {mm}^{-1}]$			0.3				
$s_{cj}=\bar{s_{cj}}+\kappa_{5\%,i}[N\cdot {mm}^{-1}]$			54.1 ± 0.3				

^1^ see Figure 13a.

**Table 7 sensors-25-04969-t007:** Stiffness of leaf spring No. 6 (j = 6). Thickness t_s_ = 2.0 mm, material—Steel.

k	j = 6						
	i	1		2		3	
	y_Mk_ [mm]	F_Mijk_ [N] ^1^	s_cijk_ [N·mm^−1^]	F_Mijk_ [N]	s_cijk_ [N·mm^−1^]	F_Mijk_ [N]	s_cijk_ [N·mm^−1^]
0	0	0	0	0	0	0	0
1	0.5	28.1	56.9	28.0	56.0	28.5	57.0
2	1	56.9	51.6	57.1	57.1	57.0	57.0
3	1.5	77.4	46.9	76.5	51.0	77.2	51.5
4	2	93.7	56.2	92.9	46.5	96.1	48.1
$\bar{s_{cjk}}=\sum_{k=1}^{5}s_{cijk}/k[N\cdot {mm}^{-1}]$			52.9	52.6		53.4	
$\bar{s_{cj}}=\sum_{i=1}^{3}\bar{s_{cijk}}/i[N\cdot {mm}^{-1}]$			53.0				
$\kappa_{5\%,i}[N\cdot {mm}^{-1}]$			0.7				
$s_{cj}=\bar{s_{cj}}+\kappa_{5\%,i}[N\cdot {mm}^{-1}]$			53.0 ± 0.7				

^1^ see Figure 13b.

**Table 9 sensors-25-04969-t009:** Effective vibration velocities v_c(x,i,j)k,0_ [mm·s^−1^] of the steel frame (i = 1) and trough (i = 2) measured in the “x” axis (l = x) of the coordinate system. Load weight m = 0 kg, leaf spring material FR4 Epoxy (* = c).

*	c (FR4 Epoxy)					
m_q_	0 kg					
i	1—Frame			${\bar{v}}_{c\left(x,1,j\right)k,0}=\sum_{j=1}^{3}v_{c\left(x,1,j\right)k,0}/j[{mm\cdot s}^{-1}]$	$\kappa_{5\%,j}[{mm\cdot s}^{-1}]$	$v_{c\left(x,1,j\right)k,0}={\bar{v}}_{c\left(x,1,j\right)k,0}+\kappa_{5\%,j}[{mm\cdot s}^{-1}]$
j	1	2	3			
k [%]	$v_{c\left(x,1,j\right)k,0}[{mm\cdot s}^{-1}]$					
10	6.6 ^1^	6.4	6.5	6.5	0.2	6.5 ± 0.2
20	18.6	18.5	18.6	18.6	0.1	18.6 ± 0.1
30	28.5	28.8	28.7	28.7	0.3	28.7 ± 0.3
40	38.1	38.8	38.6	38.5	0.6	38.5 ± 0.6
50	50.0	49.7	49.9	49.9	0.3	49.9 ± 0.3
i	2—Trough			${\bar{v}}_{c\left(x,2,j\right)k,0}=\sum_{j=1}^{3}v_{c\left(x,2,j\right)k,0}/j[{mm\cdot s}^{-1}]$	$\kappa_{5\%,j}[{mm\cdot s}^{-1}]$	$v_{c\left(x,2,j\right)k,0}={\bar{v}}_{c\left(x,2,j\right)k,0}+\kappa_{5\%,j}[{mm\cdot s}^{-1}]$
j	1	2	3			
k [%]	$v_{c\left(x,2,j\right)k,m}[{mm\cdot s}^{-1}]$					
10	9.8 ^2^	10.2	10.1	10.0	0.4	10.0 ± 0.4
20	28.6	29.4	28.8	28.9	0.7	28.9 ± 0.7
30	47.4	47.8	47.7	47.6	0.4	47.6 ± 0.4
40	64.5	65.7	65.2	65.1	1.0	65.1 ± 1.0
50	83.5	83.1	83.4	83.3	0.4	83.3 ± 0.4

^1^ see Figure 15a, ^2^ see Figure 15b.

**Table 10 sensors-25-04969-t010:** Effective vibration velocities v_c(x,i,j)k,2.57_ [mm·s^−1^] of the steel frame (i = 1) and trough (i = 2) measured in the “x” axis of the coordinate system. Load weight m = 2.57 kg, leaf spring material FR4 Epoxy (* = c).

*	c (FR4 Epoxy)					
m_q_	2.57 kg					
i	1—Frame			${\bar{v}}_{c\left(x,1,j\right)k,2.59}=\sum_{j=1}^{3}v_{c\left(x,1,j\right)k,2.59}/j$	$\kappa_{5\%,j}$	$v_{c\left(x,1,j\right)k,2.59}={\bar{v}}_{c\left(x,1,j\right)k,2.59}+\kappa_{5\%,j}$
j	1	2	3			
k	$v_{c\left(x,1,j\right)k,2.59}$					
[%]	[mm·s^−1^]					
10	10.6	10.1	10.5	10.4	0.5	10.4 ± 0.5
20	26.7 ^1^	25.9	26.5	26.4	0.7	26.4 ± 0.7
30	42.4	42.7	42.6	42.6	0.3	42.6 ± 0.3
40	56.9	56.6	56.8	56.8	0.3	56.8 ± 0.3
50	56.7	56.6	56.7	56.7	0.1	56.7 ± 0.1
i	2—Trough			${\bar{v}}_{c\left(x,2,j\right)k,2.59}=\sum_{j=1}^{3}v_{c\left(x,2,j\right)k,2.59}/j$	$\kappa_{5\%,j}$	$v_{c\left(x,2,j\right)k,m}={\bar{v}}_{c\left(x,2,j\right)k,m}+\kappa_{5\%,j}$
j	1	2	3			
k	$v_{c\left(x,2,j\right)k,2.59}$					
[%]	[mm·s^−1^]					
10	10.9	10.6	10.5	10.7	0.4	10.7 ± 0.4
20	27.4 ^1^	26.6	27.0	27.0	0.6	27.0 ± 0.6
30	42.5	43.0	43.1	42.9	0.6	42.9 ± 0.6
40	56.9	56.6	56.9	56.8	0.3	56.8 ± 0.3
50	56.9	56.6	56.8	56.8	0.3	56.8 ± 0.3

^1^ see Figure 16.

**Table 11 sensors-25-04969-t011:** Effective vibration velocities v_c(x,i,j)k,5.099_ [mm·s^−1^] of the steel frame (i = 1) and trough (i = 2) measured in the “x” axis of the coordinate system. Load weight m = 5.099 kg, leaf spring material FR4 Epoxy (* = c).

*	c (FR4 Epoxy)					
m_q_	5.099 kg					
i	1—Frame			${\bar{v}}_{c\left(x,1,j\right)k,5.099}=\sum_{j=1}^{3}v_{c\left(x,1,j\right)k,5.099}/j$	$\kappa_{5\%,j}$	$v_{c\left(x,1,j\right)k,5.099}={\bar{v}}_{c\left(x,1,j\right)k,5.099}+\kappa_{5\%,j}$
j	1	2	3			
k	$v_{c\left(x,1,j\right)k,5.099}$					
[%]	[mm·s^−1^]					
10	13.9	13.8	13.8	13.8	0.1	13.8 ± 0.1
20	23.3	22.8	23.2	23.1	0.5	23.1 ± 0.5
30	24.0 ^1^	24.3	24.2	24.2	0.3	24.2 ± 0.3
40	24.8	24.6	24.5	24.6	0.3	24.6 ± 0.3
50	24.5	24.6	24.7	24.6	0.2	24.6 ± 0.2
i	2—Trough			${\bar{v}}_{c\left(x,2,j\right)k,5.099}=\sum_{j=1}^{3}v_{c\left(x,2,j\right)k,5.099}/j$	$\kappa_{5\%,j}$	$v_{c\left(x,2,j\right)k,5.099}={\bar{v}}_{c\left(x,2,j\right)k,5.099}+\kappa_{5\%,j}$
j	1	2	3			
k	$v_{c\left(x,2,j\right)k,5.099}$					
[%]	[mm·s^−1^]					
10	6.9	7.2	7.2	7.1	0.3	7.1 ± 0.3
20	11.3	11.8	11.6	11.6	0.4	11.6 ± 0.4
30	13.3 ^1^	12.9	13.1	13.1	0.3	13.1 ± 0.3
40	12.6	12.5	12.5	12.5	0.1	12.5 ± 0.1
50	12.5	12.5	12.5	12.5	0.0	12.5 ± 0.0

^1^ see Figure 17.

**Table 12 sensors-25-04969-t012:** Effective vibration velocities v_s(x,i,j)k,0_ [mm·s^−1^] of the steel frame (i = 1) and trough (i = 2) measured in the “x” axis (l = x) of the coordinate system. Load weight m = 0 kg, leaf spring material Steel (* = s).

*	s (Steel)					
m_q_	0 kg					
i	1—Frame			${\bar{v}}_{s\left(x,1,j\right)k,0}=\sum_{j=1}^{3}v_{s\left(x1,j\right)k,0}/j[{mm\cdot s}^{-1}]$	$\kappa_{5\%,j}[{mm\cdot s}^{-1}]$	$v_{s\left(x,1,j\right)k,0}={\bar{v}}_{s\left(x,1,j\right)k,0}+\kappa_{5\%,j}[{mm\cdot s}^{-1}]$
j	1	2	3			
k [%]	$v_{s\left(x,1,j\right)k,0}[{mm\cdot s}^{-1}]$					
10	6.1	6.7 ^1^	6.3	6.4	0.5	6.4 ± 0.5
20	10.5	10.7	10.7	10.6	0.2	10.6 ± 0.2
30	10.6	10.7	10.6	10.6	0.1	10.6 ± 0.1
40	10.3	10.7	10.8	10.6	0.5	10.6 ± 0.5
50	10.3	10.7	10.7	10.6	0.4	10.6 ± 0.4
i	2—Trough			${\bar{v}}_{s\left(x,2,j\right)k,0}=\sum_{j=1}^{3}v_{s\left(x,2,j\right)k,0}/j[{mm\cdot s}^{-1}]$	$\kappa_{5\%,j}[{mm\cdot s}^{-1}]$	$v_{s\left(x,2,j\right)k,0}={\bar{v}}_{s\left(x,2,j\right)k,0}+\kappa_{5\%,j}[{mm\cdot s}^{-1}]$
j	1	2	3			
k [%]	$v_{s\left(x,2,j\right)k,0}[{mm\cdot s}^{-1}]$					
10	10.0	10.0 ^1^	10.0	10.0	0.0	10.0 ± 0.0
20	18.4	18.2	18.5	18.4	0.3	18.4 ± 0.3
30	18.0	18.1	18.4	18.2	0.4	18.2 ± 0.4
40	18.1	17.7	17.6	17.8	0.5	18.8 ± 0.5
50	18.0	17.8	17.3	17.7	0.6	17.7 ± 0.6

^1^ see Figure 18.

**Table 13 sensors-25-04969-t013:** Effective vibration velocities v_s(x,i,j)k,2.57_ [mm·s^−1^] of the steel frame (i = 1) and trough (i = 2) measured in the “x” axis (l = x) of the coordinate system. Load weight m = 2.57 kg, leaf spring material Steel (* = s).

*	s (Steel)					
m_q_	2.57 kg					
i	1—Frame			${\bar{v}}_{s\left(x,1,j\right)k,2.57}=\sum_{j=1}^{3}v_{s\left(x,1,j\right)k,2.57}/j$	$\kappa_{5\%,j}$	$v_{s\left(x,1,j\right)k,2.57}={\bar{v}}_{s\left(x,1,j\right)k,2.57}+\kappa_{5\%,j}$
j	1	2	3			
k	$v_{s\left(x,1,j\right)k,2.57}$					
[%]	[mm·s^−1^]					
10	8.7	9.2	9.1	9.0	0.5	9.0 ± 0.5
20	17.6	19.5 ^1^	19.0	18.7	1.7	18.7 ± 1.7
30	18.2	17.8	18.1	18.0	0.4	18.0 ± 0.4
40	17.9	18.6	18.4	18.3	0.6	18.3 ± 0.6
50	18.1	17.5	18.0	17.9	0.6	17.9 ± 0.6
i	2—Trough			${\bar{v}}_{s\left(x,2,j\right)k,2.57}=\sum_{j=1}^{3}v_{s\left(x,2,j\right)k,2.57}/j$	$\kappa_{5\%,j}$	$v_{s\left(x,2,j\right)k,2.57}={\bar{v}}_{s\left(x,2,j\right)k,2.57}+\kappa_{5\%,j}$
j	1	2	3			
k	$v_{s\left(x,2,j\right)k,2.57}$					
[%]	[mm·s^−1^]					
10	8.7	8.7	8.7	8.7	0.0	8.7 ± 0.0
20	19.7	22.3 ^1^	21.6	21.2	2.3	21.2 ± 2.3
30	20.0	19.6	19.9	19.8	0.4	19.8 ± 0.4
40	19.4	19.7	19.6	19.6	0.3	19.6 ± 0.3
50	19.8	19.8	19.7	19.8	0.1	19.8 ± 0.1

^1^ see Figure 19.

**Table 14 sensors-25-04969-t014:** Effective vibration velocities v_s(x,i,j)k,5.099_ [mm·s^−1^] of the steel frame (i = 1) and trough (i = 2) measured in the “x” axis (l = x) of the coordinate system. Load weight m = 5.099 kg, leaf spring material—steel (* = s).

*	s (Steel)					
m_q_	5.099 kg					
i	1—Frame			${\bar{v}}_{s\left(x,1,j\right)k,5.099}=\sum_{j=1}^{3}v_{s\left(x,1,j\right)k,5.099}/j$	$\kappa_{5\%,j}$	$v_{s\left(x,1,j\right)k,5.099}={\bar{v}}_{s\left(x,1,j\right)k,5.099}+\kappa_{5\%,j}$
j	1	2	3			
k	$v_{s\left(x,1,j\right)k,5.099}$					
[%]	[mm·s^−1^]					
10	7.5	7.6	7.6	7.6	0.1	7.6 ± 0.1
20	6.7	8.5	7.9	7.7	1.6	7.7 ± 1.6
30	6.8	6.7 ^1^	6.7	6.7	0.1	6.7 ± 0.1
40	7.1	7.0	7.1	7.1	0.1	7.1 ± 0.1
50	6.8	7.5	7.3	7.2	0.6	7.2 ± 0.6
i	2—Trough			${\bar{v}}_{s\left(x,2,j\right)k,5.099}=\sum_{j=1}^{3}v_{s\left(x,2,j\right)k,5.099}/j$	$\kappa_{5\%,j}$	$v_{s\left(x,2,j\right)k,5.099}={\bar{v}}_{s\left(x,2,j\right)k,5.099}+\kappa_{5\%,j}$
j	1	2	3			
k	$v_{s\left(x,2,j\right)k,5.099}$					
[%]	[mm·s^−1^]					
10	16.1	15.9	16.2	16.1	0.3	16.1 ± 0.3
20	19.0	18.4	18.6	18.7	0.5	18.7 ± 0.5
30	19.1	19.0 ^1^	19.2	19.1	0.2	19.1 ± 0.2
40	18.9	19.0	19.0	19.0	0.1	19.0 ± 0.1
50	19.0	19.3	19.2	19.2	0.3	19.2 ± 0.3

^1^ see Figure 20.

**Table 15 sensors-25-04969-t015:** Effective vibration velocities v_p(x,i,j)k,0_ [mm·s^−1^] of the steel frame (i = 1) and trough (i = 2) measured in the “x” axis (l = x) of the coordinate system. Load weight m = 0 kg, leaf spring material—plastic (* = p).

*	p (Plastic PCCF)					
m_q_	0 kg					
i	1—Frame			${\bar{v}}_{p\left(x,1,j\right)k,0}=\sum_{j=1}^{3}v_{p\left(x,1,j\right)k,0}/j[{mm\cdot s}^{-1}]$	$\kappa_{5\%,j}[{mm\cdot s}^{-1}]$	$v_{p\left(x,1,j\right)k,0}={\bar{v}}_{p\left(x,1,j\right)k,0}+\kappa_{5\%,j}[{mm\cdot s}^{-1}]$
j	1	2	3			
k [%]	$v_{p\left(x,1,j\right)k,0}[{mm\cdot s}^{-1}]$					
10	6.4	6.5	6.5	6.5	0.1	6.5 ± 0.1
20	18.9	19.1	19.1	19.0	0.2	19.0 ± 0.2
30	32.1	32.5	32.4	32.3	0.4	32.3 ± 0.4
i	2—Trough			${\bar{v}}_{p\left(x,2,j\right)k,0}=\sum_{j=1}^{3}v_{p\left(x,2,j\right)k,0}/j[{mm\cdot s}^{-1}]$	$\kappa_{5\%,j}[{mm\cdot s}^{-1}]$	$v_{p\left(x,2,j\right)k,0}={\bar{v}}_{p\left(x,2,j\right)k,m}+\kappa_{5\%,j}[{mm\cdot s}^{-1}]$
j	1	2	3			
k [%]	$v_{p\left(x,2,j\right)k,0}[{mm\cdot s}^{-1}]$					
10	10.6	10.9	10.8	10.8	0.3	10.8 ± 0.3
20	31.2	31.3	31.2	31.2	0.1	31.2 ± 0.1
30	52.8	53.4	53.1	53.1	0.5	53.1 ± 0.5

**Table 16 sensors-25-04969-t016:** Effective vibration velocities v_p(x,i,j)k,2.57_ [mm·s^−1^] of the steel frame (i = 1) and trough (i = 2) measured in the “x” axis (l = x) of the coordinate system. Load weight m = 2.57 kg, leaf spring material—plastic (* = p).

*	p (Plastic PCCF)					
m_q_	2.57 kg					
i	1—Frame			${\bar{v}}_{p\left(x,1,j\right)k,2.57}=\sum_{j=1}^{3}v_{p\left(x,1,j\right)k,2.57}/j$	$\kappa_{5\%,j}$	$v_{p\left(x,1,j\right)k,2.57}={\bar{v}}_{p\left(x,1,j\right)k,2.57}+\kappa_{5\%,j}$
j	1	2	3			
k	$v_{p\left(x,1,j\right)k,2.57}$					
[%]	[mm·s^−1^]					
10	9.7	9.7	9.7	9.7	0.0	9.7 ± 0.0
20	28.6	27.2	27.9	27.9	1.1	27.9 ± 1.1
i	2—Trough			${\bar{v}}_{p\left(x,2,j\right)k,2.57}=\sum_{j=1}^{3}v_{p\left(x,2,j\right)k,2.57}/j$	$\kappa_{5\%,j}$	$v_{p\left(x,2,j\right)k,2.57}={\bar{v}}_{p\left(x,2,j\right)k,2.57}+\kappa_{5\%,j}$
j	1	2	3			
k	$v_{p\left(x,2,j\right)k,2.57}$					
[%]	[mm·s^−1^]					
10	10.2	10.2	10.2	10.2	0.0	10.2 ± 0.0
20	29.2	28.1	28.8	28.7	0.9	28.7 ± 0.9

**Table 17 sensors-25-04969-t017:** Effective vibration velocities v_p(x,i,j)k,5.099_ [mm·s^−1^] of the steel frame (i = 1) and trough (i = 2) measured in the “x” axis (l = x) of the coordinate system. Load weight m = 5.099 kg, leaf spring material—plastic (* = p).

*	p (Plastic PCCF)					
m_q_	5.099 kg					
i	1—Frame			${\bar{v}}_{p\left(x,1,j\right)k,5.099}=\sum_{j=1}^{3}v_{p\left(x,1,j\right)k,5.099}/j$	$\kappa_{5\%,j}$	$v_{p\left(x,1,j\right)k,5.099}={\bar{v}}_{p\left(x,1,j\right)k,5.099}+\kappa_{5\%,j}$
j	1	2	3			
k	$v_{p\left(x,1,j\right)k,5.099}$					
[%]	[mm·s^−1^]					
10	6.4	6.1	6.4	6.3	0.3	6.3 ± 0.3
20	20.4	21.1	20.6	20.7	0.6	20.7 ± 0.6
i	2—Trough			${\bar{v}}_{p\left(x,2,j\right)k,5.099}=\sum_{j=1}^{3}v_{p\left(x,2,j\right)k,5.099}/j$	$\kappa_{5\%,j}$	$v_{p\left(x,2,j\right)k,5.099}={\bar{v}}_{p\left(x,2,j\right)k,5.099}+\kappa_{5\%,j}$
j	1	2	3			
k	$v_{p\left(x,2,j\right)k,5.099}$					
[%]	[mm·s^−1^]					
10	11.4	11.0	11.3	11.2	0.4	11.2 ± 0.4
20	31.9	34.6	33.4	33.3	2.2	33.3 ± 2.2

## Data Availability

Measured data of effective vibration speed values v_*(l,i,j)k,m_ [mm·s^−1^], listed from Table 9, Table 10, Table 11, Table 12, Table 13, Table 14, Table 15, Table 16 and Table 17 and processed using DEWESoftX software, can be sent in case of interest, by prior written agreement, in * DXD (DewesoftX Data File) format or * XLSX (Microsoft Excel) * XLSX (Microsoft Excel) format.

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
