# Peer review of "Using Acceleration Sensors to Diagnose the Operating Condition and to Detect Vibrating Feeder Faults"

_sensors, 2025, doi:10.3390/s25164969_

Round 1
Reviewer 1 Report
Comments and Suggestions for Authors
Dear authors
I have overall enjoyed article reading. The topic discussed by the authors is interesting to readers, but there are multiple flaws in English grammar that must be attended before further article processing. I list below some major and minor corrections that should be considered as well:
Major corrections
Introduction section: This section must be reorganized and split into subsections. This sections spans between lines 34 and 155, and discuss multiple topics such as: previous works, fundamental physics and current work; therefore, subsection splitting is required for a better comprehension.
Also Introduction section: What is the aim of this work? Please make sure to clearly state what the purpose of this article is before jumping to section 2.
Line 158: The actual size of what? the size of prototype shown in Figure 1? Or did you mean the magnitude of the listed elements in lines 159 through 163? Please avoid stray sentences as they lead to misunderstandings.
Line 175: What is a frame 4?
Figure 8: The text within Figure 8(c) is blurry and barely readable. Please reorganize this figure so that text within figure 8(c) is more readable.
Figures 9 and 10: There are so much information and lines within these figures that I could not get the message. Please reconsider image splitting and/or rearranging.
Results section: It was hard for me to follow up the results reported in this section. There are multiple graphs and tables that demonstrate the effort placed by the authors, but also, they are not properly organized and presented. A list or sketch that summarizes in advance the experimental test of this section may help the readers to understand the methodology behind all the measurements; the list/sketch could be presented at the beginning of section 3, so that the readers may be aware of what’s coming.
Minor corrections
Lines 65-68: What means to “directly depend on” and to “indirectly depend on”? It is not clear if this comparison is about direct/inverse proportionality or something else.
Lines 70-86: These lines discussed about previous results in specialized literature. However, when first reading these lines, it was hard to get the overall picture behind electrical current, number of coils, and so on. Finally, in line 86 the authors introduced Figure 1 which depicts the experimental setup initially described in lines 70-75. Please rewrite this subsection so that the indication to figure 1 appears at the beginning.
Equation (2): Similar to the observation from “lines 70-86”, equation (2) was formally introduced in line 93, but it was originally discussed in lines 66-67.
Line 101: Parenthesis is not terminated “)”.
Table 8: what does they symbol “ž” stand for in the sentence “j = 7 až 10”?
Comments on the Quality of English LanguageLine 19: Considering that allow is a transitive verb, the pronoun “us” should be used after the verb, i.e. “that allow us”
Line 26: The following sentence is just too large, please consider rewriting: “We can state that from the analysis of the measured vibration velocities transmitted to the steel frame of the vibrating feeders, it is possible to….”
Line 39: the preposition “on” should be added before whether, i.e. “on whether the monitoring...”
Line 48: Considering you are citing an already published study, the verb analyze should be written in past tense.
Line 65: This sentence should be written as “directly depends on”
Line 75: Similar to the correction from line 48, the verb “present” should be written in past tense.
Also line 75: The preposition “a” should be used before “theoretical modelling”
Author Response
Dear Reviewer,
Thank you very much for your valuable advice and comments, which have helped to improve the quality of our manuscript.
We would like to respond to your questions in the attached DOCX file.
We kindly ask you to review our responses and to show understanding for any shortcomings in addressing your comments.
On behalf of all co-authors,
L. Hrabovský.

Reviewer 2 Report
Comments and Suggestions for Authors
In this manuscript, the authors propose “Using acceleration sensors to diagnose the operating condition and to detect vibrating feeder faults.” Overall, the paper addresses a specific engineering problem; however, as an academic contribution its writing quality is inadequate. Moreover, the application and innovation of the theoretical methods offer little to engage the reader. Consequently, the manuscript requires substantial revision. Specific comments are as follows:
- The layout of the paper needs improvement. For instance, the paragraphing in the Introduction and in the Experimental section is inappropriate.
- The Introduction contains an excessive review of tangential literature. In addition, the relevance of the equations presented in the Introduction to the present study is unclear. I recommend that the authors briefly summarize the main content and key contributions of the research in the Introduction.
- The figures are cluttered; many contain large amounts of extraneous information.
- Given that the core of this manuscript is to investigate the feasibility of using vibration signals for fault diagnosis, I strongly urge the authors to provide an installation diagram showing exactly where and how the vibration sensors are mounted; this is critical to the experimental validity.
- In the Experimental section, how were the actual faults simulated? Please clarify the authors’ rationale for selecting and implementing each fault scenario.
- In the experiments, is varying only the spring stiffness and load mass sufficient to represent system faults? The authors should justify whether these two parameters alone can adequately reflect real-world fault conditions.
- The references are outdated; I recommend that the authors incorporate the most recent literature to ensure the work is situated within the current state of the art.
- I suggest shortening the Conclusions, as the current version includes substantial irrelevant material.
- The manuscript contains linguistic and typographical errors.
Author Response
Dear Reviewer,
thank you for the time you have dedicated to reviewing our manuscript. I also sincerely appreciate your valuable suggestions and comments, which will help improve the quality of our work.
In the attached DOCX file, we have attempted to address your questions and to revise or remove parts of the manuscript that were not appropriately written.
We kindly ask you to review our responses and to show understanding for any shortcomings in addressing your comments.
On behalf of all co-authors,
L. Hrabovský.

Reviewer 3 Report
Comments and Suggestions for Authors
The study is technically interesting and has potential. Below is the list of my comments and for the manuscript.
- The Introduction does not articulate what is new beyond prior work that also used frame mounted accelerometers to infer feeder load or fault state. Explicitly state why existing frame level vibration metrics are inadequate, how your 3 sensor–1 feeder setup overcomes this, and what quantitative advance (error, latency, cost) you deliver.
- Ten pages derive basic electromagnet and spring formulas that are textbook knowledge; most are never used in later analysis. Move derivations to an appendix or delete; instead, add a concise model that links measured velocity to trough mass and spring stiffness, then validate that model experimentally.
- Environmental conditions (temperature, mounting torque, power supply voltage) are absent; these strongly affect electromagnetic force.
- The authors claim “remote diagnostics” but there are no data transmission, cloud processing, or decision logic is demonstrated; you only log data locally and “state that it can be sent remotely”. Either remove “remote” from title/claims or add a telemetry experiment with latency and packet loss statistics.
- The Discussion mainly restates that velocity changes with mass (obvious); you must explain why FR4 springs invert the frame to trough ratio at 2.57 kg but steel does not.
Author Response
Dear Reviewer,
thank you for the time you have dedicated to reviewing our manuscript. I also sincerely appreciate your valuable suggestions and comments, which will help improve the quality of our work.
In the attached PDF file, we have attempted to address your questions and to revise or remove parts of the manuscript that were not appropriately written.
We kindly ask you to review our responses and to show understanding for any shortcomings in addressing your comments.
On behalf of all co-authors,
L. Hrabovský.

Reviewer 4 Report
Comments and Suggestions for Authors
- The experimental methods and ideas of the manuscript are relatively novel, but there are major problems in the structure of the content and the layout of the charts and graphs are expected to be corrected by the authors, specifically as follows
- Section 3 mentioned in the No. 1 steel plate and No. 3 steel plate thickness, measurement of physical quantities, experimental conditions, experimental materials are the same, in the completion of the No. 1 plate of the stiffness of the experiments, why repeat the No. 3 plate of the experiments, and the results of the two experiments are also different, and the same No. 2 and No. 4 steel plate the same problem exists.
- There is a lack of description of a number of graphs and tables, including tables 3, 4, 5, 6, and 7, the presentation of the experimental content of the manuscript is incomplete, and there are serious problems with the overall structure.
- There is a mismatch between the graphic analysis, the charts do not match the analytical text about the charts, which makes them difficult to read, and there are serious problems with the layout of the manuscript and the design of the placement of the charts.
Author Response

(The authors gave the same response as above.)

Round 2
Reviewer 1 Report
Comments and Suggestions for Authors
Dear authors
The manuscript has greatly improved from its original version. Nonetheless, there are still some formatting and organization considerations that must be improved before further article processing; they are below listed:
Introduction Section
As previously mentioned in the first review, this section discussed about literature review, fundamental physics, current developments within the field and aims of the paper; all this was discussed without adding any subsection to “1. Introduction”; please divide the introduction into subsections “1.1, 1.2, and so on”, so that reading this section becomes easier.
Also in the introduction section: Bear in mind that already published works mut be cited in past tense, e.g. Line 54 “described”, Line 105 “declared”, Line 138 “analyzed”.
Comments on the Quality of English LanguageAlso in the introduction section: Bear in mind that already published works mut be cited in past tense, e.g. Line 54 “described”, Line 105 “declared”, Line 138 “analyzed”.
Author Response
Dear Reviewer.
on behalf of all co-authors, I would like to sincerely thank you for your recommendations and valuable comments, which have contributed to the improvement of our manuscript.
We also respectfully appreciate your suggestion to correct the inappropriately used verbs. We are very pleased that you accepted our previous responses.
I would also like to express my gratitude for your time and for all your valuable advice and suggestions.
We kindly provide our responses to your comments in the attached DOCX file.
Once again, we thank you respectfully.
We wish you continued success.
Sincerely,
L. Hrabovský

Reviewer 2 Report
Comments and Suggestions for Authors
Accept
Author Response
Dear Reviewer.
I would like to sincerely thank you (on behalf of all co-authors) for your recommendation to publish our manuscript in the journal Sensors. We are very pleased that you have accepted our previous comments. I would also like to thank you for your time and for all your valuable advice and suggestions.
Once again, we extend our respectful thanks.
We wish you continued success. Yours sincerely, L. Hrabovský.
Reviewer 3 Report
Comments and Suggestions for Authors
Thanks to the authors for the response, I have no further comments.
Author Response
Dear Reviewer.
We are very pleased that you accepted our previous comments. I would like to sincerely thank you for recommending our manuscript for publication in the journal Sensors. We greatly appreciate your review and thank you for everything.
With respect and warm regards,
Leopold Hrabovský.
Reviewer 4 Report
Comments and Suggestions for Authors
The question has been modified, there are no other questions, agreed to accept
Author Response
Allow me to sincerely thank you for your valuable feedback. I truly appreciate your comments and we are very grateful that you recommended our article for publication in the journal Sensors.
I wish you much success in your work.
With kind regards,
Leopold Hrabovský